# DexGarmentLab: Dexterous Garment Manipulation Environment with Generalizable Policy

**Yuran Wang[1*]**   **Ruihai Wu[1*]**   **Yue Chen[1*]**
Jiarui Wang[1]   Jiaqi Liang[1]   Ziyu Zhu[1]   Haoran Geng[2]
Jitendra Malik[2]   Pieter Abbeel[2]   Hao Dong[1†]
[1]Peking University   [2]University of California, Berkeley

## Abstract

Garment manipulation is a critical challenge due to the diversity in garment categories, geometries, and deformations. Despite this, humans can effortlessly handle garments, thanks to the dexterity of our hands. However, existing research in the field has struggled to replicate this level of dexterity, primarily hindered by the lack of realistic simulations of dexterous garment manipulation. Therefore, we propose **DexGarmentLab**, the first environment specifically designed for dexterous (especially bimanual) garment manipulation, which features large-scale high-quality 3D assets for 15 task scenarios, and refines simulation techniques tailored for garment modeling to reduce the sim-to-real gap. Previous data collection typically relies on teleoperation or training expert reinforcement learning (RL) policies, which are labor-intensive and inefficient. In this paper, we leverage garment structural correspondence to automatically generate a dataset with diverse trajectories using only a single expert demonstration, significantly reducing manual intervention. However, even extensive demonstrations cannot cover the infinite states of garments, which necessitates the exploration of new algorithms. To improve generalization across diverse garment shapes and deformations, we propose a Hierarchical gArment-manipuLation pOlicy (HALO). It first identifies transferable affordance points to accurately locate the manipulation area, then generates generalizable trajectories to complete the task. Through extensive experiments and detailed analysis of our method and baseline, we demonstrate that HALO consistently outperforms existing methods, successfully generalizing to previously unseen instances even with significant variations in shape and deformation where others fail. Our project page is available at: https://wayrise.github.io/DexGarmentLab/.

## 1   Introduction

The ability to manipulate various objects is critical for general robots. Despite advancements in the manipulation of rigid [12] and articulated [33] objects, deformable objects, and garments in particular, continue to pose substantial challenges [34, 36, 35] due to their highly variable geometries and intricate deformations. Despite this, humans handle garments with remarkable ease using dexterous hands, thanks to their superior adaptability, larger manipulation area, and coordinated finger control, highlighting the importance of equipping robots with similar capabilities. Dexterous (especially bimanual) hands enable stable and precise actions (such as catching, cradling, pinching, and smoothening, as shown in Fig. 1), and excel in complex tasks like tie knotting and assisted dressing, where multi-finger coordination ensures accurate and adaptive manipulation.

Dexterous garment manipulation faces three key challenges: (i) *Data*: The high-dimensional action space of dexterous hands and the complex nature of garments make policy learning data-intensive [14, 6]. What's more, different garment manipulation tasks require different hand grasp poses to ensure smooth manipulation and make garments maintain the desired condition. Directly collecting real-world data is impractical due to high cost. Thus, researchers have pursued simulators for garment manipulation [23, 26, 39, 16, 25], but these simulators often rely on labor-intensive teleoperation

39th Conference on Neural Information Processing Systems (NeurIPS 2025).

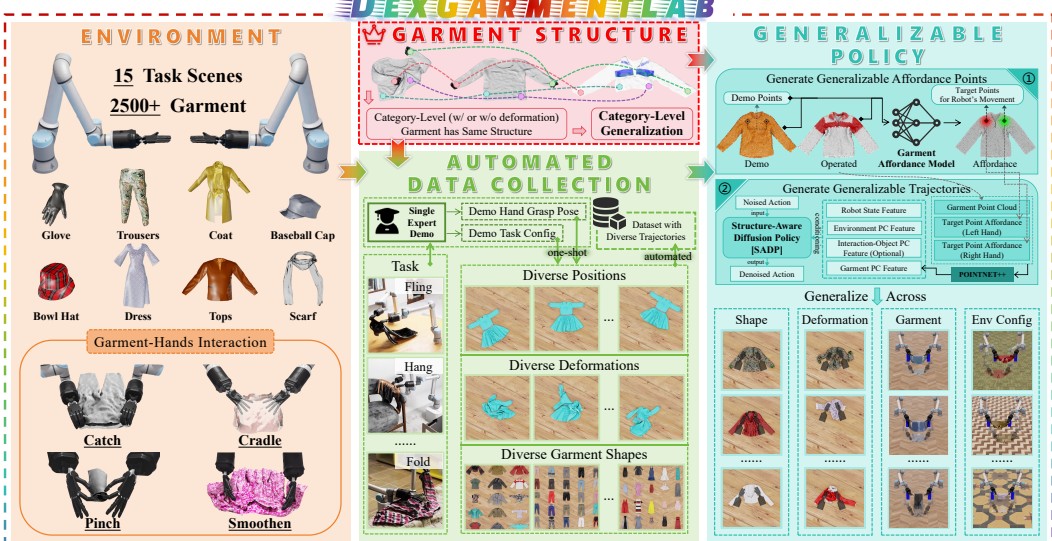

Figure 1: **Overview.** DexGarmentLab includes three major components: **Environment**, **Automated Data Collection** and **Generalizable Policy**. Firstly, we propose Dexterous Garment Manipulation Environment with 15 different task scenes (especially for bimanual coordination) based on 2500+ garments. Because of the same structure of category-level garment, category-level generalization is accessible, which empowers our proposed automated data collection pipeline to handle different position, deformation and shapes of garment with task config (including grasp position and task sequence) and grasp hand pose provided by single expert demonstration. With diverse collected demonstration data, we introduce **H**ierarchical g**A**rment manipu**L**ation p**O**licy (**HALO**), combining affordance points and trajectories to generalize across different attributes in different tasks.

or expert RL policies to collect demonstrations, which are inefficient. (ii) *Environment*: Effective garment manipulation involves interactions not only with garments but also with rigid and articulated objects (such as hangers, the human body, etc.). Current simulators [23, 39] lack the necessary level of realism and physical accuracy for complex interactions, particularly with dexterous hands. (iii) *Algorithm*: Garment manipulation requires understanding diverse geometries, complex states, and difficult goals. Existing reinforcement learning (RL) [44, 23] or imitation learning [1, 6] approaches often require intricate task-specific reward designs or extensive demonstrations, limiting their scalability to real-world applications.

To address these challenges, we introduce **DexGarmentLab** (Fig. 1), an environment *built upon Isaac Sim* and specially designed for dexterous (especially bimanual) garment manipulation, featuring: (1) **Diverse and Realistic Environments**: A large-scale dataset of more than 2,500 garments in 8 categories from ClothesNet [45], high-quality 3D assets for 15 task scenarios, paired with advanced simulation techniques to reduce sim-to-real gap. (2) **Automated Data Collection Pipeline**: An automated pipeline that generates diverse demonstrations based on garment structural correspondence with the help of a single expert demonstration, facilitating the generation of large and varied datasets without requiring manual intervention. (3) **Generalizable Policy**: a **H**ierarchical g**A**rment-manipu**L**ation p**O**licy (**HALO**) which leverages affordance (for locating garment manipulation areas) and diffusion method (for generating trajectories based on garment and scene), achieving better generalization in garment manipulation than previous imitation learning algorithms.

In summary, our contributions include:

- We introduce **DexGarmentLab Environment**, the first simulation environment for dexterous (especially bimanual) garment manipulation, featuring a wide range of task scenarios, high-quality assets, and realistic physical interactions.

- We propose a pipeline for **automated data collection**, generating diverse demonstrations in various task scenarios and reducing the need for manual intervention.

- We propose **H**ierarchical g**A**rment-manipu**L**ation p**O**licy (**HALO**), a novel **hierarchical** framework that leverages affordance and diffusion method to enable generalizable manipulation of diverse garments.

- **Extensive experiments and detailed analysis** of our approach and baseline in both simulation and real-world settings, demonstrating its data efficiency and generalization ability significantly outperforming baseline methods.

## 2 Related Work

### 2.1 Deformable Object Simulation

Current deformable simulations [18, 23, 39, 21] are limited in the types of objects they support and lack realistic physical interactions, which hinders dexterous garment manipulation research. For instance, softgym [23] is confined to simulating tops and trousers while fluidlab [39] can only simulate fluids. Although Lu et al. [25] extends to various object types, it relies on attaching invisible cubes to garments, failing to simulate realistic physical interactions. In contrast, we introduce DexGarmentLab, which incorporates extra physical parameters (adhesion, friction, particle-adhesion/friction-scale, etc.) to guarantee stable and realistic interaction between garments and robots. Additionally, simulators [23, 26, 39, 16, 25] often rely on labor-intensive teleoperation or expert RL policies to collect demonstrations, which are inefficient. Thus, we propose an automated pipeline that exploits structural correspondence across garment categories to collect demonstrations, eliminating manual effort. Table 1 shows comparisons between DexGarmentLab and other environments.

Table 1: Comparisons with Other Deformable Object Environments

| Simulator | Scene | Garment | Sim on GPU | Multiple Dexhands | DexHand Garment Task | Data Collection Method | Demonstration Data | Physically Plausible Robot-Garment Interaction |
|---|---|---|---|---|---|---|---|---|
| Softgym | ✗ | ✓ | ✓ | ✗ | ✗ | Manual | ✗ | ✗ |
| PyBullet | ✗ | ✓ | ✗ | ✗ | ✗ | Manual | ✗ | ✗ |
| Fluidlab | ✗ | ✗ | ✓ | ✗ | ✗ | Manual | ✗ | ✗ |
| Dexterous Gym | ✗ | ✗ | ✗ | ✗ | ✗ | Manual | ✗ | ✗ |
| Sapien | ✓ | ✗ | ✓ | ✓ | ✗ | Manual | ✗ | ✗ |
| GarmentLab | ✓ | ✓ | ✓ | ✗ | ✗ | Manual | ✗ | ✗ |
| **DexGarmentLab** | ✓ | ✓ | ✓ | ✓ | ✓ | Automated | ✓ | ✓ |

### 2.2 Dexterous Manipulation

Dexterous manipulation has emerged as a critical research frontier, with applications in tasks like in-hand manipulation [40, 15, 29, 37, 8], articulated object manipulation [4, 19, 9, 42] and deformable object manipulation [21, 38, 43]. However, most researches focus on single-handed manipulation, overlooking bimanual dexterity, which is essential for tasks like cradling garments and pinching gloves (Shown in Fig. 1). While recent efforts have demonstrated bimanual capabilities in specialized contexts including lid manipulation [22], dynamic object interception [17], and articulated object manipulation [42], these approaches remain fundamentally limited to rigid-body dynamics. Our work performs the first investigation on the learning-based bimanual dexterous manipulation of garment and build the pioneering environment with diverse scenarios covering different garment categories.

### 2.3 Garment and Deformable Object Manipulation

While much research has focused on manipulating simple deformable objects like square-shaped cloths [34, 38, 24], ropes and cables [31, 34, 38], and bags [2, 7], garment manipulation presents a substantial challenge. Garment manipulation involve diverse geometries, complex deformations, and fine-grained actions. Many existing studies on dexterous garment manipulation rely on optimize method [3, 13], which struggles with the high freedom of bimanual dexterous hands. Zhaole et al. [44], Lin et al. [23] attempt to solve bimanual dexterous manipulation tasks with reinforcement learning (RL). However, they require intricate reward designs tailored to specific manually designed tasks. Avigal et al. [1], Canberk et al. [6] rely on large-scale annotated data, which is labor-intensive and time-consuming, hindering the scalability in the scenarios of real-world applications. In this paper, we introduce **H**ierarchical g**A**rment-manipu**L**ation p**O**licy (**HALO**) which leverages affordance and diffusion method to facilitate manipulating diverse unseen category-level garments in multiple tasks with different scene configurations.

## 3 DexGarmentLab Environment

In this section, we present the construction of DexGarmentLab, the first environment specifically designed for dexterous (especially bimanual) garment manipulation and built upon IsaacSim 4.5.0.

### 3.1 DexGarmentLab Environment Setup

**Observation Space.** The observation space includes both proprioceptive and visual data. Proprioceptive data comprises robot joint positions, end-effector 6D poses, and other kinematic information.

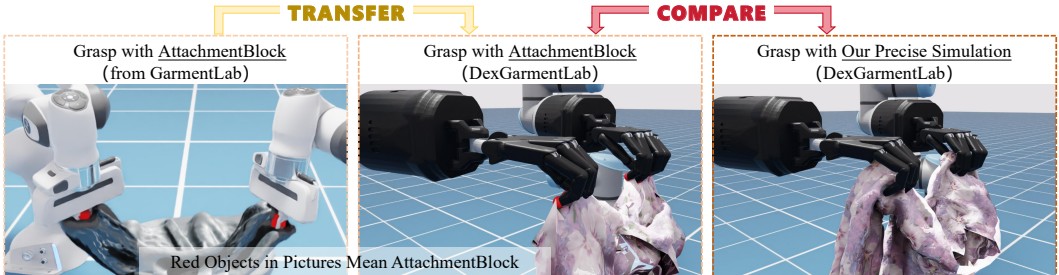

Figure 2: **Comparison of Garment-Robot Interactions between GarmentLab and DexGarment-Lab.** *Left*: In GarmentLab, Franka grasp garment with red block attached. *Middle*: We transfer *AttachementBlock* method to dexterous hands and set red block at the tip of each finger (ten blocks totally). The performance is not so good, as described in 3.2. *Right*: Our method (DexGarmentLab) can make the interactions between dexterous hands and garment more natural.

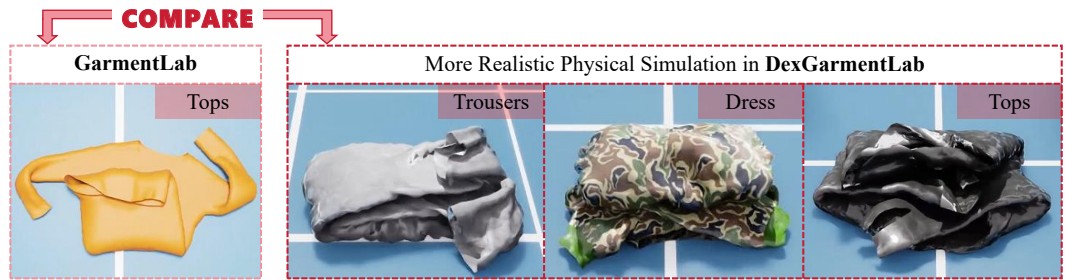

Figure 3: **Compare Garment Properties between GarmentLab and DexGarmentLab.** *Left*: After folding in GarmentLab, the garment struggles to maintain a stable folded state and easily becomes disorganized. *Right*: With realistic physical simulation in DexGarmentLab, folded garments can stably maintain their folded states, which means Garment Properties is more mature and natural.

Visual data includes point clouds captured by depth camera and RGB images. The point cloud is cropped to the robot's workspace and down-sampled for efficiency.

**Action Space.** The action space is a 60-dimensional vector: 6 DoF for each arm and 24 DoF for each Shadow Hand. The UR10e arms can be controlled using both Inverse-Kinematics (IK) Controller given end-effector 6D poses and Proportional-Derivative (PD) Controller given joint positions, while Shadow Hands are controlled using PD controller based on joint positions.

## 3.2 DexGarmentLab Physcial Simulation

**Simulation Method.** To achieve realistic simulation, we employ methods tailored to the physical properties of garments. Large garments (e.g., tops, dresses, trousers, etc.) are simulated using Position-Based Dynamics (PBD) [27], while small, elastic items (e.g., gloves, hats) are modeled via the Finite Element Method (FEM) [5]. We provide detailed introduction and selection reason about PBD and FEM in Appendix I.1. Human avatars are represented by articulated skeletons with rotational joints and a skinned mesh for lifelike rendering.

**Key Design for Physical Garment Simulation.** PBD is widely used for simulating most garments, but its loosely connected particles often allow grippers to penetrate the garment without achieving effective lifting. GarmentLab introduces attach blocks to address this, enabling garment-gripper attachment (Fig. 2, left). However, this approach fails to capture realistic interactions, resulting in unnatural sagging when applied to dexterous hands (Fig. 2, middle). Moreover, even minimal contact—such as a single finger block touching the garment—can establish attachment and lift the garment, which is clearly unreasonable.

Therefore, we introduce adhesion (*between particle and rigid*), friction (*between particle and rigid*) and particle-scale (*between particles*) parameters to enhance realism. Benefiting from friction and adhesion, dexhands can grasp and lift garments based on physical force without attach blocks (Fig. 2, right), while particle-adhesion (or -friction)-scale stabilizes the particle system, preventing excessive self-collisions between particles which cause garments to become disorganized (Fig. 3). We provide more details in Appendix A.

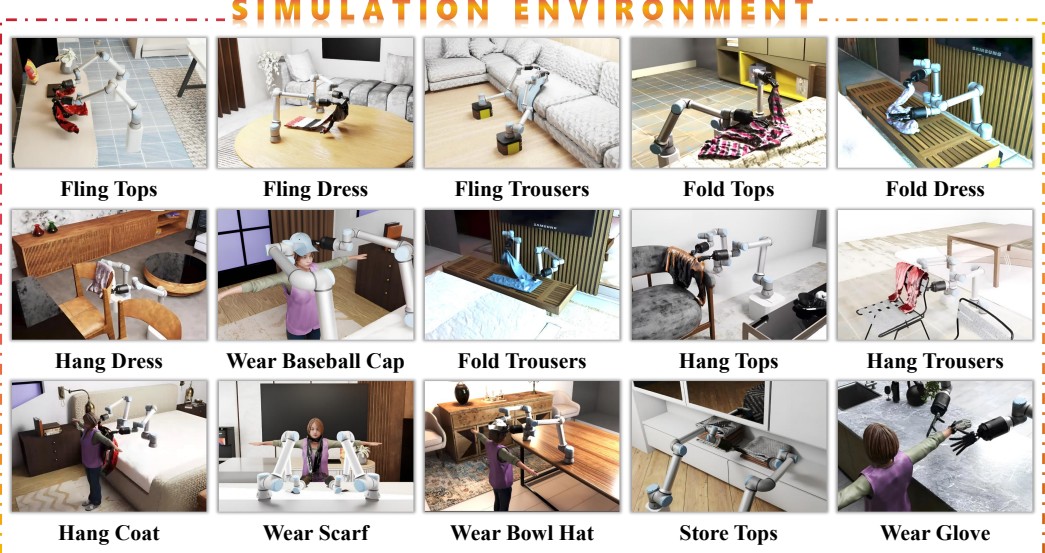

Figure 4: **DexGarmentLab Simulation Environment.** We introduce 15 garment manipulation tasks across 8 categories, encompassing both garment-self-interaction (e.g., Fling, Fold) and garment-environment-interaction (e.g., Hang, Wear, Store) scenarios. In garment-self-interaction tasks, key variables include garment position, orientation, and shape. In garment-environment-interaction tasks, environment-interaction assets positions (e.g., hangers, pothooks, humans, etc.) are also considered.

### 3.3 Asset Selection and Annotation

We use garment models from the ClothesNet dataset [45], which contains over 2,500 garments across 8 categories (e.g., tops, coats, trousers, dresses, etc.), and build environment-interaction assets (such as hangers, pothooks, humans, etc.). We provide plain meshes customizable with colors and textures for garments to support both realistic and controlled experimental setups. Controlled randomness in placement for both garments and environment-interaction assets—through limited rotations and translations—maintains task feasibility while enhancing generalization in policy learning.

### 3.4 DexGarmentLab Tasks

Dexterous (especially bimanual) garment manipulation is vital for domestic applications, yet it has not been thoroughly explored in existing research. To address this, we introduce 15 tasks across 8 garment categories (Fig. 4). Further details on these tasks are available in Appendix K.

## 4 Automated Data Collection

Collecting data through teleoperation or RL is highly labor-intensive, especially for dexterous garment manipulation tasks, due to the diverse shapes and deformations of garments and the high-dimensional action space of dexterous hands. This makes automated data collection essential, with the key challenges being: 1) identifying appropriate manipulation points across different garment configurations, and 2) generating task-specific hand poses accordingly.

In our proposed automated data collection pipeline, for a given task, we begin with a single expert demonstration to extract key information: hand grasp poses, task sequences, and demo grasp points on the garment. Leveraging the Garment Affordance Model (Refer Sec. 4.1), we use affordance to identify target grasp points on novel garments with diverse deformations corresponding to demo grasp points. Then, the pipeline executes the task sequence based on inferred points and hand grasp poses, thereby enabling efficient and scalable data collection. Sec. 4.2 explains the whole procedure.

### 4.1 Garment Affordance Model (GAM)

Built upon the UniGarmentManip [35] framework, GAM leverages structural and correspondence consistency across category-level garments, enabling the identification of target grasp points on category-level novel garments. For training process, as shown in Fig. 5 (Red Part), we employ a **Skeleton Merger** [32] network architecture to obtain the skeleton point correspondences between flat garments while adopting the point tracing method in simulation to establish correspondences between the flat garment and its deformed version. Using InfoNCE [20] loss function, we train PointNet++ [28] to pull features of positive corresponding point correspondences closer while pushing apart negative corresponding pairs, which enhances dense visual correspondence by enabling alignment across different garments in various states. Please refer [35] for more details.

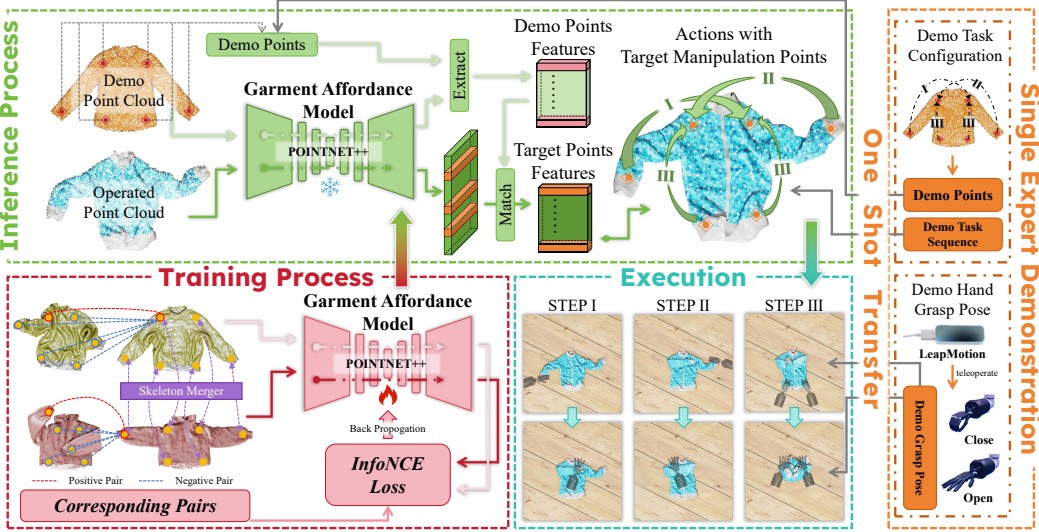

Figure 5: **Automated Data Collection Pipeline.** Given a single expert demonstration, we can get demo points, demo task sequences and demo grasp poses for the specific task. Category-level garment (w/ or w/o deformation) has almost the same structure, base on which we can train Garment Affordance Model (GAM) with category-level generalization. With GAM (refer Sec. 4.1), we match demo points from the demo garment point cloud $O$ to a new garment point cloud $O'$ and control robot to execute the specific task based on the demo task sequences (through trajectory retargeting) with dexhands' movement guided by demo hand grasp poses (through PD controller based on joint positions). 'Fold Tops' task is shown as example in this figure.

It's worth mentioning that, to enable GAM to handle point clouds with translation and scale invariance, we pre-normalize the input demo/operated point cloud into a canonical space. This ensures that GAM maintains generalization ability when faced with garments undergoing different translations and scales. As for the issue of rotational invariance, we consider it as the garment's deformation state. By generating sufficient samples with varying rotations, we enable GAM to effectively learn the correspondences of garments under different rotations.

For inference process, as shown in Fig. 5 (Green Part), with pretrained GAM, given one demo garment point cloud $O$, demo grasp points $(p_1, p_2, ...)$ and one operated garment point cloud $O'$, we can obtain the demo grasp points features $(f_{p_1}, f_{p_2}, ...)$ and operated garment observation features. By dotting demonstration grasp points features and operated garment observation features to get similarities and selecting features with biggest similarity scores, we can get corresponding grasp points $(p'_1, p'_2, ...)$ on $O'$.

## 4.2 Automated Data Collection Pipeline

Here we give a detailed description about our automated data collection pipeline shown in Fig. 5.

Firstly, We obtain demo grasp points, demo task sequences, and demo hand grasp poses from a single expert demonstration. In our actual operation process, while grasp points and task sequences are manually defined, hand poses are generated using the *LeapMotion* solution (see Appendix B).

Once target grasp points are identified on the operated garment using GAM, with demo task sequences, we control the robotic arms via inverse kinematics (IK) to execute sequential operations while controlling dexterous hands using PD controller based on the joint positions from demo hand grasp poses. It is important to note that the task trajectories are not fixed but are adapted based on the garment's shape and length. For example, in *garment-self-interaction tasks*, such as folding, the lifting height is adjusted according to the sleeves and overall garment length to ensure proper folding. In *garment-environment-interaction tasks*, such as hanging, both the lifting height and placement position are adapted to align the garment's center with the hanger, preventing slippage. These adjustments reflect common and reasonable actions in real-world scenarios, and introducing such variations increases the task difficulty. The details about tasks can be found in Appendix K.

During task execution, we can simultaneously record various information (such as images, point clouds, robot joint states, etc.) within the simulation environment. These serve as expert demonstration data for subsequent offline training of the policy. Details about recorded information can be found in Appendix D.

# 5 Generalizable Policy

When dealing with garments, which exhibit highly complex deformation states, current mainstream imitation learning (IL) algorithms (e.g. Diffusion Policy [10], Diffusion Policy 3D [41]) show relatively poor generalization (as evidenced by our experimental results shown in Tab. 2). The main issue is that IL-based trajectories fail to accurately reach the target manipulation points on garments with new shapes and deformations, while also being unable to generate suitable trajectories based on the garment's own shape and structure, ultimately leading to manipulation failures.

To address this, we propose **H**ierarchical g**A**rment manipu**L**ation p**O**licy (**HALO**), a generalizable policy to solve the manipulation of garments with complex deformations and uncertain states. **HALO** is decomposed into two major stages, as shown in Fig. 6

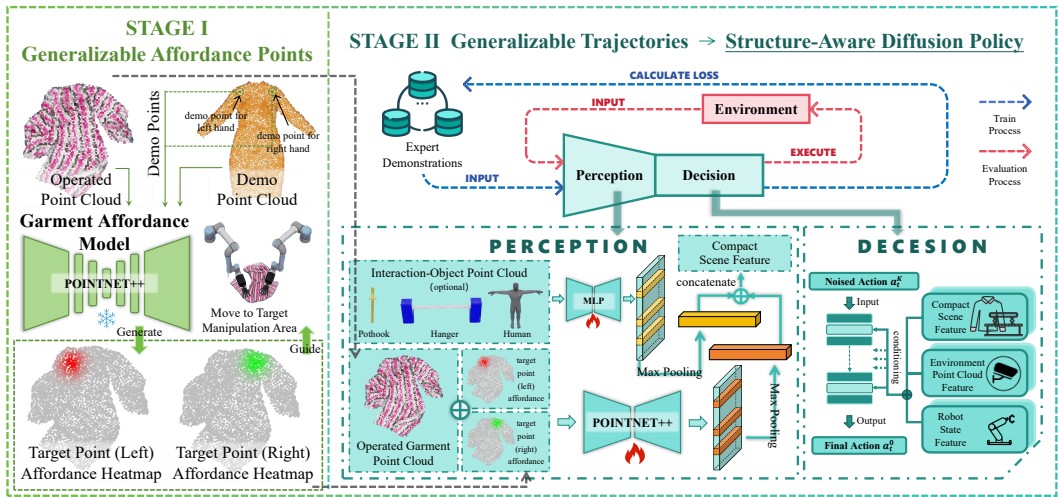

Figure 6: **Generalizable Policy.** We adopt hierarchical structure to implement Generalizable Policy. Firstly, we use GAM to generate generalizable affordance points, which will be used for robots to locate and move to target area. Secondly, we introduce Structure-Aware Diffusion Policy (SADP), which extracts features from garment point cloud (with left and right point affordances as binding features), interaction-object point cloud, environment point cloud and robot joint states as condition to generate joint actions (including 24 DOF for each hand and 6 DOF for each arm, totally 60 DOF).

In the first stage, we use GAM to accurately locate the manipulation area of the garment, addressing the limitation of previous IL in grasping brand-new clothes at the correct position. Refer Sec. 4.1 for GAM's details. We next focus on the design of the Structure-Aware Diffusion Policy (SADP).

Due to the poor generalization ability of current mainstream methods such as DP and DP3 for complex and variable garment manipulation scenarios, we propose **Structure-Aware Diffusion Policy (SADP)**, a garment-environment-generalizable diffusion policy that improves the generalization for different garment shapes and scene configurations, thereby enabling the smooth generation of subsequent trajectories after moving to the target manipulation area guided by GAM.

SADP fundamentally follows the framework of Diffusion Policy [10], with the primary distinction lying in its observation representation, denoted as $s$, which is elaborated below.

With operated garment point cloud and left / right target point affordances generated by GAM, we concatenate them together and use PointNet++ [28] to extract garment feature $F_{garment}$, while using MLP-based Feature Extractors to extract interaction-object feature $F_{object}$. $F_{garment}$ and $F_{object}$ are concatenated into a compact scene feature $F_{scene}$. At each timestep, the full environment point cloud $O_{environment}$ and the robot state $O_{state}$ are encoded using MLP and fused with $F_{scene}$ to form the denoising condition $s$ for SADP. As for Garment-Self-Interaction tasks without interaction-object point cloud, we only use $F_{garment}$ to be $F_{scene}$, which means interaction-object point cloud is optional. Here, $F_{garment}$ captures current garment state (position, shapes, structure, etc.), while $F_{object}$ reflects current interaction-object state (position, etc.).

Through experimental validation, SADP exhibits better generalization capabilities. We will further illustrate this advantage with experimental results in Sec. 6.2. Training details can be found in Appendix H.

# 6 Experiment

## 6.1 Environment Setup

**Tasks and Environments.** We evaluate our method on 14 garment manipulation tasks with varying deformation characteristics. Detailed environment specifications including scene randomization parameters, success metrics, and train/test configurations are provided in the Appendix K.

**Demonstration Collection.** Using our automated data collection pipeline (Sec. 4), we acquired 100 demonstrations per task, with 30-100 environment steps in each demonstration. The time required to collect a demonstration varies between approximately 30-80 seconds depending on the task, which is significantly faster than data collection through teleoperation. We make comparison between autonomously collected data and teleoperation data in Appendix F.

**Baselines.** We compare against two state-of-the-art diffusion-based approaches across all the garment manipulation tasks: 1) **Diffusion Policy (DP)** [10], which utilizes images as observations to generate actions via diffusion. 2) **3D Diffusion Policy (DP3)** [41], which replaces image inputs in the Diffusion Policy with point clouds. What's more, we have also additionally included four new baselines for comparison: **ACT (IL), pi0 (VLA), RDT (VLA), and Eureka (RL+VLM)** and select representative tasks for evaluation, including Fling Dress, Fold Trousers, Hang Coat, and Wear Bowlhat in simulation, as well as Fold Tops in real world, which can be found in Appendix G.

**Ablations.** To analyze components, we evaluate two ablated variants: 1) **w/o GAM**: Excludes the Garment Affordance Model (GAM), using SADP for trajectory execution. 2) **w/o SADP**: Removes Structure-Aware Diffusion Policy (SADP), using GAM + DP3 for trajectory execution.

**Metrics.** Each task is evaluated over 50 episodes with three different seeds. We report success rates as $Mean \pm Std$ across all trials.

Table 2: **Quantitative Results in Simulation. HALO** outperforms baselines and ablations.

| Method | Fling | | | Fold | | | Hang | | | | Wear | | | Store |
|---|---|---|---|---|---|---|---|---|---|---|---|---|---|---|
| | Dress | Tops | Trousers | Dress | Tops | Trousers | Dress | Tops | Trousers | Coat | Baseball Cap | Bowlhat | Scarf | Tops |
| DP | 0.59 ± 0.05 | 0.55 ± 0.05 | 0.54 ± 0.15 | 0.55 ± 0.03 | 0.53 ± 0.05 | 0.47 ± 0.04 | 0.51 ± 0.03 | 0.45 ± 0.01 | 0.56 ± 0.09 | 0.52 ± 0.04 | 0.65 ± 0.03 | 0.41 ± 0.05 | 0.67 ± 0.07 | 0.65 ± 0.02 |
| DP3 | 0.51 ± 0.03 | 0.54 ± 0.03 | 0.58 ± 0.10 | 0.47 ± 0.08 | 0.52 ± 0.06 | 0.54 ± 0.07 | 0.51 ± 0.08 | 0.53 ± 0.09 | 0.59 ± 0.08 | 0.58 ± 0.04 | 0.61 ± 0.02 | 0.55 ± 0.04 | 0.60 ± 0.01 | 0.63 ± 0.08 |
| Ours w/o GAM | 0.66 ± 0.02 | 0.68 ± 0.06 | 0.71 ± 0.08 | 0.61 ± 0.04 | 0.65 ± 0.02 | 0.62 ± 0.10 | 0.71 ± 0.03 | 0.64 ± 0.02 | 0.75 ± 0.07 | 0.62 ± 0.01 | 0.64 ± 0.02 | 0.62 ± 0.03 | 0.66 ± 0.02 | 0.70 ± 0.01 |
| Ours w/o SADP | 0.68 ± 0.09 | 0.67 ± 0.07 | 0.68 ± 0.09 | 0.55 ± 0.08 | 0.53 ± 0.09 | 0.63 ± 0.08 | 0.69 ± 0.10 | 0.70 ± 0.08 | 0.68 ± 0.06 | 0.71 ± 0.03 | 0.70 ± 0.01 | 0.64 ± 0.02 | 0.64 ± 0.05 | 0.58 ± 0.07 |
| Ours (HALO) | **0.82 ± 0.06** | **0.85 ± 0.05** | **0.83 ± 0.02** | **0.76 ± 0.02** | **0.81 ± 0.03** | **0.77 ± 0.02** | **0.88 ± 0.05** | **0.92 ± 0.04** | **0.91 ± 0.02** | **0.90 ± 0.01** | **0.79 ± 0.03** | **0.72 ± 0.04** | **0.88 ± 0.01** | **0.80 ± 0.02** |

## 6.2 Results Analysis

Tab. 2 (Row 1, 2, 5) quantifies **HALO**'s performance against baseline methods in simulation. Our method achieves superior success rates across dexterous garment manipulation tasks, demonstrating statistically significant improvements over existing approaches. We also provide more baselines comparison in Appendix G.

Our ablation study (Tab. 2, Row 3, 4, 5) quantifies the impact of ablating two core components in **HALO**: the Garment Affordance Model (GAM) and the Structure-Aware Diffusion Policy (SADP). Fig. 7 provides visual evidence of these effects.

Through Tab. 2, we find that the performance of HALO markedly decreases when **GAM** is excluded. The integration of GAM leverages dense visual correspondence, thereby enhancing the model's performance to locate precise manipulation area (shown in Fig. 7), particularly for tasks with significant variability in garment shapes and deformations.

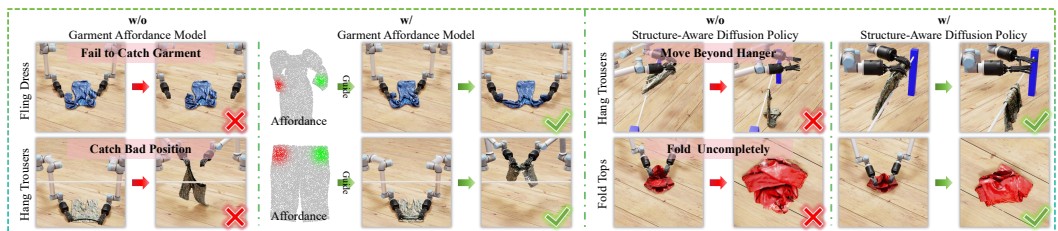

Figure 7: **Effects brought by GAM / SADP. Left:** Without GAM, robots fail to catch the right manipulation points. **Right:** Without SADP, robots fail to adjust the trajectories based on garment shapes and structures.

Moreover, **SADP** substantially boosts the performance of HALO. When the model encounters garments that differ considerably from those in the training set, **SADP** can assist in **adjusting** the trajectories according to the garments' own shapes and structures, thereby better accomplishing

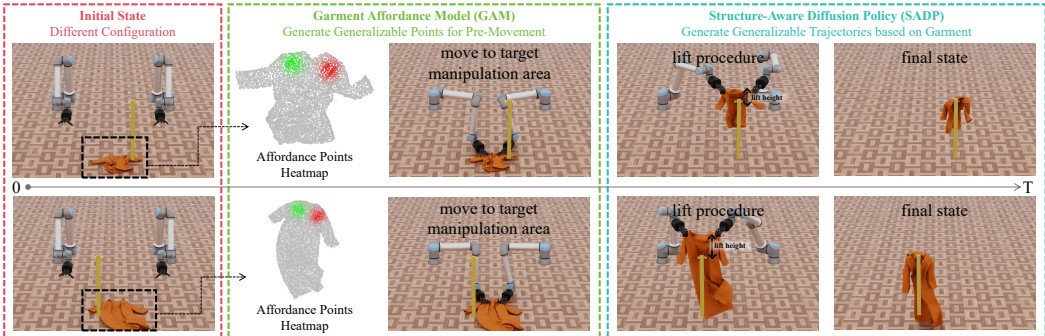

Figure 8: **HALO's whole procedure.** Using "Hang Coat" as an example. GAM first infers target manipulation points for robot's movement, enabling generalization across different garments. Then, SADP generates trajectories based on the garment and scene configurations. Despite variations in garment shape, length, and pothook positions across scenes, SADP adapts accordingly, moving to accurate positions and lifting coats to appropriate heights to successfully hang them on the pothook.

the tasks. For instance, as shown in Fig. 7, in the "Hang Trousers" task, the shape of the trousers determines the appropriate hanging height and forward-moving distance; the HALO can adjust its trajectory to complete the task. In the "Fold Tops" task, the HALO can adjust the folding position according to the garment structure, making the folds neater.

**HALO** enables precise manipulation point inference and generates robust policies for diverse garment and scene configurations. Figure 8 further illustrates this.

### 6.3 Real-World Experiments

There are two ways using HALO for garment manipulation tasks:

1) **transfer the data collection pipeline from simulation to real world, conduct automated data collection and policy training directly in the real world.**

2) **train the policy in the simulation and transfer the policy to the real world.**

#### 6.3.1 Experiments and Results Analysis for Way 1

we adopt **way 1** to demonstrate the effectiveness of HALO. In the Appendix F Tab. 5, we report the efficiency and success rate of real-world data collection, highlighting the strong sim-to-real performance of GAM.

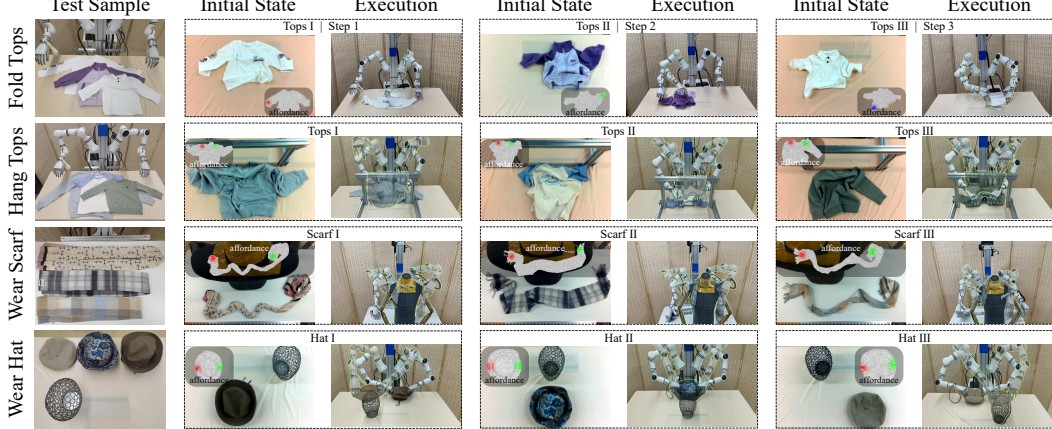

Figure 9: **Real-World Experiments (Way 1).** We select Fold_Tops (Garment-Self-Interaction), Hang_Tops, Wear_Scarf, Wear_Hat (Garment-Environment-Interaction) as typical tasks for illustration. The test samples have different shapes, length, deformations while position of garments and interaction objects are variable. Despite this, GAM of HALO ensures accurate manipulation area location for grasp while SADP of HALO ensures adaptive trajectories base on garment and interaction object.

**Setup.** Our setup comprises two RealMan RM75-6F (Arms) with Psibot G0-R (Dexterous Hands) and a RealSense D435 camera (As shown in Appendix E). We use Segment-Anything-2 [30] to segment the garment and interaction object from the scene and obtain corresponding point clouds.

Table 3: **Real-World Evaluation** on different tasks.

| Task | Fold Tops | Hang Tops | Wear Scarf | Wear Hat |
|------|-----------|-----------|------------|----------|
| DP | 9 / 15 | 10 / 15 | 6 / 15 | 10 / 15 |
| DP3 | 8 / 15 | 8 / 15 | 7 / 15 | 9 / 15 |
| Ours(HALO) | **13 / 15** | **13 / 15** | **11 / 15** | **14 / 15** |

**Evaluation.** We evaluate our proposed method on 4 tasks: Fold Tops, Hang Tops, Wear Scarf, and Wear Hat. For each task, we have 3 distinct garments per category, each with 5 initial deformations. Shown in Tab. 3, our method outperforms all baselines. Fig. 9 demonstrates the excellent performance of our proposed method.

### 6.3.2 Experiments and Results Analysis for Way 2

Sim-to-real transfer from the simulation environment remains an important aspect of our study. To this end, as shown in Fig. 10, we aligned the settings of both the simulation and real-world environments by using the same hardware setup—Shadow Hand and UR10e—and selected two tasks, Hang Trousers and Wear Hat, for policy-level sim-to-real transfer, which means way 2. The evaluation criteria follow those used in the previous real-world experiments.

It is worth noting that sim-to-real performance is more sensitive to point cloud noise. To address the limited precision of the Realsense D435 in this context, we employed a Kinect camera for more accurate point cloud acquisition.

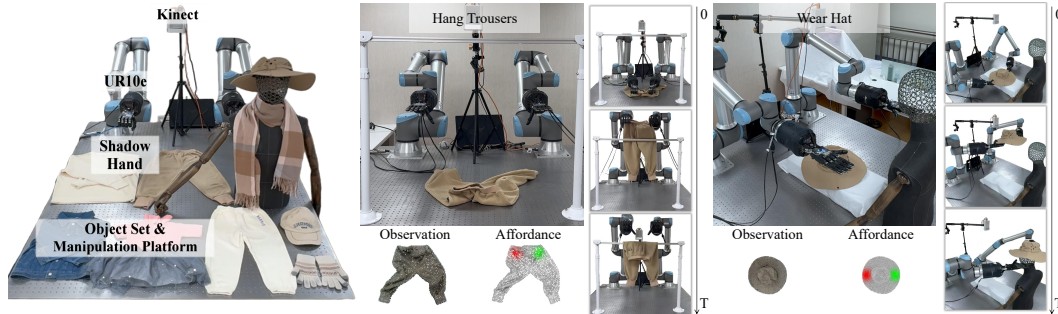

Figure 10: **Real-World Experiments (Way 2).** Two UR10e paired with ShadowHands and an Azure Kinect camera are used in our sim-to-real experiment. For each task, we show the scene configuration, affordance generated by GAM and trajectories generated by SADP.

Table 4: Performance Impact of Adding Real-World Data to Simulation Data

| Task Name | Only Simulation Data | Simulation Data + 15 Real-World Data |
|-----------|----------------------|--------------------------------------|
| Hang Trousers | 8 / 15 (53.3%) | 13 / 15 (86.7%) |
| Wear Hat | 9 / 15 (60.0%) | 13 / 15 (86.7%) |

Experimental results in Tab. 4 show that due to the gap between simulation and the real world, training the policy solely on simulated data leads to a drop in sim-to-real performance. Incorporating a small amount of real-world data into the training process can effectively enhance the policy's generalization ability.

## 7 Conclusion

In this paper, we introduce **DexGarmentLab**, the first simulation environment designed to address the challenges of dexterous (especially bimanual) garment manipulation. Our work mainly makes three key contributions, including *DexGarmentLab Environment*, *Automated Data Collection* and *Generalizable Policy*. Through extensive experiments, we demonstrate that our approach can effectively learn complex manipulation tasks with minimal supervision and generalize across a wide range of garment shapes and deformation states in both simulation and real-world environments. The limitation of our work is discussed in Appendix I.

## Acknowledgment

This project was supported by National Natural Science Foundation of China (62376006) and National Youth Talent Support Program (8200800081).

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

# Appendix Overview

## A Details about Key Parameters in PBD for Improved Realism

Comprehensive explanation about how the parameters—*adhesion*, *friction*, *particle-adhesion-scale*, *particle-friction-scale*—contribute to improving the realism of garment in the simulation environment.

## B LeapMotion for Teleoperation

Detailed overview of the LeapMotion workflow and its operational performance.

## C Advantages of Dexterous Hands over Parallel Grippers

Detailed explanation on advantages of dexterous hands over parallel grippers.

## D Recorded Information in Automated Data Collection

Comprehensive introduction to the various types of environmental information collected in the simulation during the automated data collection phase, along with the techniques employed for efficient data acquisition.

## E Real-World Experiment Scene

Schematic Diagram of the Real-World Experimental Setup.

## F Comparison with Teleoperation Data

Detailed comparison between autonomously collected data and teleoperation data to demonstrate the efficiency and high quality of automated data collection pipeline.

## G Additional Baseline Comparison

Additional baseline comparison for demonstrating the excellent performance of HALO.

## H Training Details of Main Algorithm

Training details for the GAM and SADP algorithms, including hyperparameter settings, computational resources, and other relevant specifications.

## I Limitation

The limitation of our work, including simulation-method limitation, task limitation, data-collection limitation.

## J DexGarmentLab Assets

A detailed description of the various assets used in DexGarmentLab, including garment assets, environment-interaction assets, robots assets, and material assets.

## K Detailed Task Description

A detailed description of all garment manipulation tasks involved in DexGarmentLab, including task environment initialization and randomization, task sequence, task success metrics, and garment assets used for each task.

## L Broader Impact

The potential societal impacts of our work.

# A  Details about Key Parameters in PBD for Improved Realism

Position-Based Dynamics (PBD) is a widely adopted method for simulating deformable objects such as garment. It operates by enforcing geometric constraints directly on particle positions, offering both numerical stability and computational efficiency. In this section, we elaborate on several key parameters — *adhesion*, *friction*, *particle-adhesion-scale*, and *particle-friction-scale* — which significantly influence the realism and stability of garment-object interactions, particularly in dexterous manipulation scenarios.

## A.1  Adhesion

Adhesion introduces artificial attractive forces between particles or between particles and surfaces (e.g., a robotic hand). It is used to simulate surface stickiness, enabling persistent contact during manipulation. Although adhesion is not a physical force in classical mechanics, it can be heuristically modeled as:

$$\mathbf{f}_{\text{adh}} = -k_{\text{adh}} \cdot (\mathbf{x}_i - \mathbf{x}_j), \quad \text{if } \|\mathbf{x}_i - \mathbf{x}_j\| < r_{\text{adh}}, \tag{1}$$

where $\mathbf{x}_i$ and $\mathbf{x}_j$ are the positions of two particles (or a particle and a surface proxy), $k_{\text{adh}}$ is the adhesion coefficient, and $r_{\text{adh}}$ is the adhesion radius threshold.

**Effect:** Adhesion enables garments to maintain contact with the fingers of a dexterous hand without requiring continuous high-pressure gripping, facilitating reliable grasping and lifting.

## A.2  Friction

Friction resists relative tangential motion and is critical for grasp stability. The classical Coulomb friction model is expressed as:

$$\mathbf{f}_{\text{fric}} = -\mu \cdot \|\mathbf{f}_n\| \cdot \frac{\mathbf{v}_t}{\|\mathbf{v}_t\|}, \tag{2}$$

where $\mu$ is the friction coefficient, $\mathbf{f}_n$ is the contact normal force, and $\mathbf{v}_t$ is the tangential relative velocity. In PBD, friction is often approximated as a positional correction that opposes sliding during constraint projection steps.

**Effect:** Friction allows garment to resist sliding off hand surfaces, enhancing control during manipulation.

## A.3  Particle-Adhesion-Scale

This parameter scales the adhesion forces between internal garment particles. It helps prevent excessive separation or instability during self-collisions or folding. High values increase inter-particle attraction, leading to more cohesive motion.

**Effect:** Particle-adhesion-scale improves garment stability by preventing explosive separations or chaotic folding, especially when complex self-contact occurs.

## A.4  Particle-Friction-Scale

Particle-friction-scale controls the internal friction between garment particles during relative motion. It is applied during internal constraint solving (e.g., stretch, shear, or collision resolution) and acts as a damping mechanism:

**Effect:** This parameter suppresses excessive internal sliding, preserving folds and wrinkles, and enabling more physically plausible draping and manipulation behaviors.

## A.5  Summary

Together, these parameters significantly enhance the realism of garment manipulation in simulation. Benefiting from adhesion and friction, dexterous hands can grasp and lift garments based on contact

forces. Meanwhile, the particle-adhesion and friction scales stabilize internal garment dynamics, reducing self-collision artifacts and preserving structural coherence.

## B    LeapMotion for Teleoperation

We employ the ***Leap Motion Controller*** as a teleoperation device to control the Shadow Hand and generate task-specific hand poses, thereby facilitating the automated data collection pipeline. The Leap Motion Controller is a compact USB device designed for desktop use. It leverages two 640×240-pixel near-infrared cameras to capture hand motion within a roughly hemispherical interaction volume extending up to approximately 60 cm, typically operating at 120 Hz. Its internal algorithms process the raw spatial data to extract 27 distinct hand features, including the palm normal vector, hand direction, wrist position, and 24 finger joint positions.

Here we just utilize the 24 finger joint positions to control Shadow Hand's movement. The teleoperation system and visualization performance are shown in Fig 11.

We have provided relevant tutorial about how to use LeapMotion in our released code.We also explain details on how LeapMotion generates grasping poses below.

For single specific task category (e.g., Hang Coat), the hand poses required to perform specific actions in specific regions, can generally be treated as consistent. Moreover, the deformable nature of garments allows them to adapt naturally to the dexterous hand's pose during manipulation. Therefore, for each specific task, we first define a set of task-specific hand poses (the number depends on the task; for example, Hang Coat requires both a closed grasping pose for the collar and an open pose for release). These poses are generated via teleoperation using LeapMotion, with physical feasibility considered during generation. In this way, the pose definition process inherently includes a manual filtering step to select the most suitable poses.

These task-specific hand poses are then transferred directly into the data collection process for that task category. That is, during data collection across different garments within the same task, the same predefined hand poses are reused.

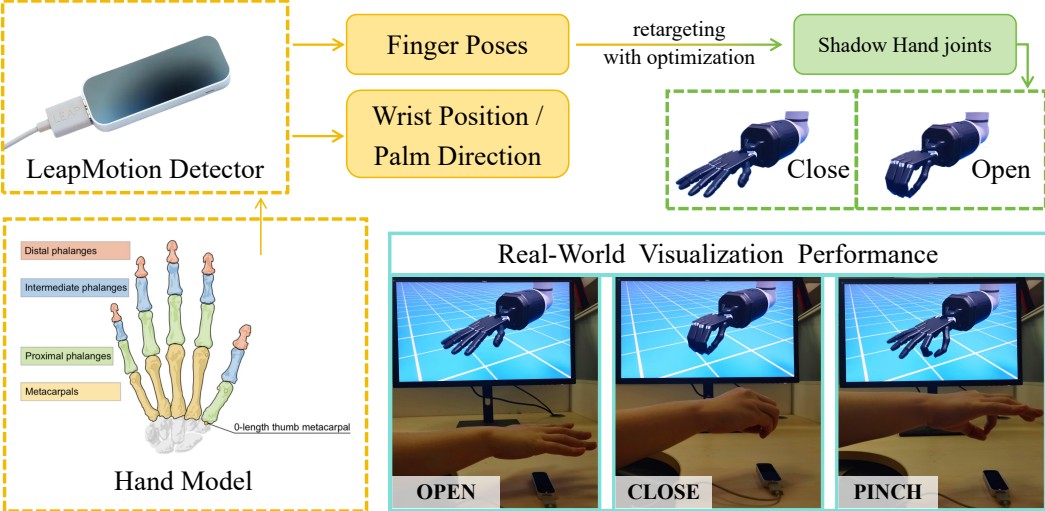

Figure 11: **LeapMotion Workflow and Real-World Visualization Performance.**

## C    Advantages of Dexterous Hands over Parallel Grippers

Due to current restrictions in simulation technology, certain complex tasks cannot yet be effectively implemented in simulated environments. However, across the existing tasks, dexterous hands still demonstrate clear advantages over parallel grippers in several aspects.

1. Our experimental results, particularly those real-world results, reveal significant limitations in two-finger grippers when handling thin or flat regions of garments. In such cases, **the gripper frequently**

**fails to achieve a stable grasp, with garments either slipping during manipulation or not being grasped at all**. Many existing demonstrations involve garment folding with two-finger grippers rely on grasping the garment's boundary, but these strategies also often leads to repeated failed attempts. In contrast, our system employing a five-finger dexterous hand demonstrates clear advantages. **Owing to the hand's larger operation range, greater surface contact, and coordinated multi-finger control, it can stably grasp various regions of a garment—not just its edges—and maintain a firm hold throughout the manipulation process**. This capability greatly enhances the flexibility and robustness of robotic garment handling, freeing the system from constraints on graspable areas or predefined strategies.

2. Furthermore, **the diverse hand postures offered by the dexterous hand allow us to tailor the manipulation strategy to the specific requirements of each task**.

- to accomplish a fine-grained task such as putting on a glove, the hand can adopt a **pinch** posture to precisely grasp the glove edge.

- For stable garment pick-and-place operations, an **open/close** posture can be used to maximize contact and grip.

- When placing folded clothes, a **cradle** posture helps preserve the folded structure.

- To smooth wrinkles, the hand can switch to a **smoothening** pose that maximizes surface coverage.

These example poses are shown in the bottom-left corner of Figure 1 (main paper).

The dexterous hand not only replicates the functionality of two-finger grippers but also enables finer and more robust manipulation of garments.

3. Although current simulation still faces significant challenges in accurately modeling interactions between deformable objects and dexterous hands—limiting its ability to simulate complex tasks—these limitations are expected to be gradually addressed with the continued advancement of computer graphics and robotics technologies. As a result, **the advantages of dexterous hands will become even more prominent in complex tasks such as assistive dressing or knot tying, which require flexible coordination among multiple fingers**. We will continue to extend tasks and optimize the simulation framework based on DexGarmentLab, aiming to support a wider range of more complex dexterous-hand garment interaction tasks.

## D   Recorded Information in Automated Data Collection

In this section, we provide a detailed overview of the various types of data collected during our automated data collection process. These include:

- **joint_state**: The joint values of the dual-arm dexterous system, including the left arm and right arm (6 joints each), left hand and right hand (24 joints each). The shape is (60, ).

- **image**: RGB images of the workspace. The shape is (480, 640, 3).

- **env_point_cloud**: Point cloud of the workspace with color, downsampled to 2048. The shape is (2048, 6).

- **garment_point_cloud**: Point cloud of the garment without color, downsampled to 2048. The shape is (2048, 3).

- **points_affordance_feature**: The left / right point affordance feature generated by GAM, which can be seen as similarity score (normalized to [0,1]). The shape is (2048, 2).

- **object_point_cloud**: Point cloud of the environment-interaction object without color, downsampled to 2048, only exist in Garment-Environment-Interaction tasks. The shape is (2048, 3).

It is worth noting that during data collection, we record all the aforementioned information **every five time steps**. Empirical validation shows that this approach not only reduces data density and accelerates policy training, but also does not compromise overall performance during validation.

## E  Real-World Experiment Scene

Our real-world experiment scene comprises RealMan RM75-6F (Arms) with Psibot G0-R (Dexterous Hands), a RealSense D435 camer and a few garments across different categories, which is shown in Fig. 12.

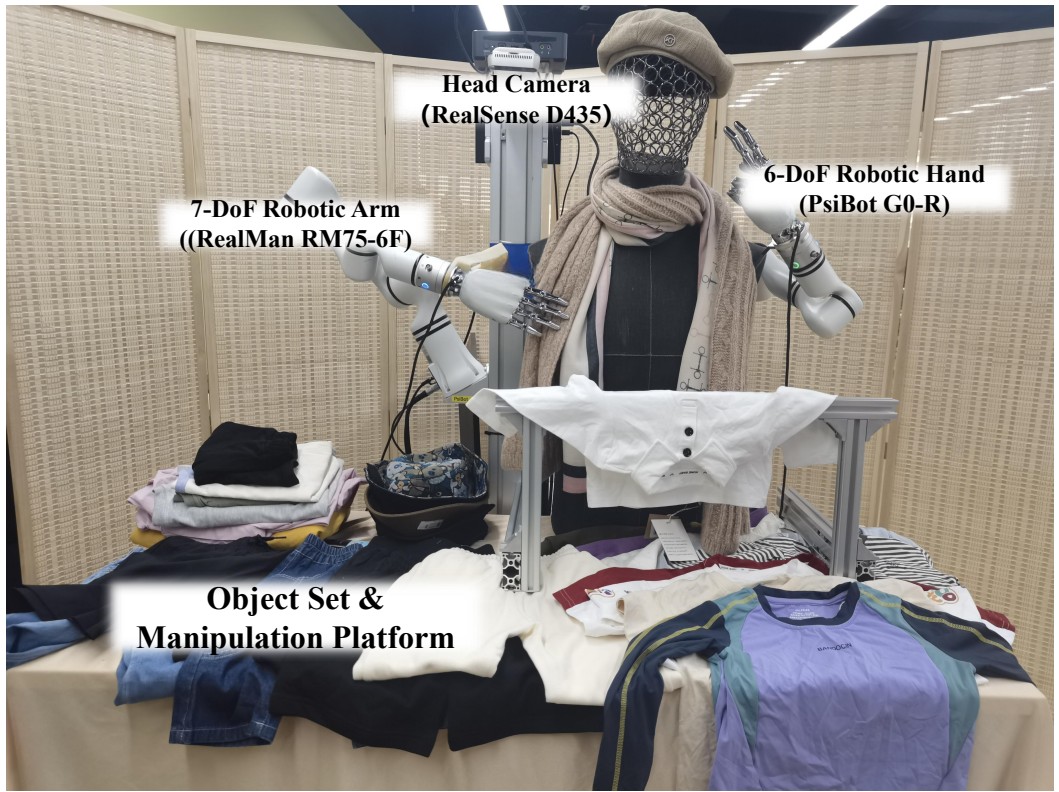

Figure 12: **Real-World Experiment Scene**

## F  Comparison with Teleoperation Data

Table 5: **Automated Data Collection Time** and **Success Rate** Across All Tasks

| Task Name | Collection Success Rate | Single Collection Time | Task Name | Collection Success Rate | Single Collection Time |
|---|---|---|---|---|---|
| Fling Dress | 92.6% (100/108) | 0 min 56 s | Hang Trousers | 99.0% (100/101) | 0 min 58 s |
| Fling Tops | 90.1% (100/111) | 0 min 55 s | Store Tops | 98.0% (100/102) | 1 min 05 s |
| Fling Trousers | 90.1% (100/111) | 0 min 48 s | Wear Baseball Cap | 85.5% (100/117) | 0 min 55 s |
| Fold Dress | 84.7% (100/118) | 1 min 06 s | Wear Scarf | 83.3% (100/120) | 1 min 42 s |
| Fold Tops | 82.0% (100/122) | 1 min 08 s | Wear Bowl Hat | 95.2% (100/105) | 0 min 53 s |
| Fold Trousers | 82.6% (100/121) | 0 min 56 s | Fold Tops (Real-World) | 90.9% (50/55) | 0 min 50 s |
| Hang Coats | 92.6% (100/108) | 0 min 41 s | Hang Tops (Real-World) | 96.2% (50/51) | 0 min 38 s |
| Hang Dress | 90.9% (100/110) | 0 min 48 s | Wear Scarf (Real-World) | 83.3% (50/60) | 1 min 15 s |
| Hang Tops | 91.7% (100/109) | 0 min 46 s | Wear Hat (Real-World) | 93.8% (50/53) | 0 min 36 s |

Tab. 5 presents the data collection time and success rate across all tasks, which validates the effectiveness and efficiency of the automated data collection pipeline. To further verify the quality of data acquired via automation, three representative tasks were selected for data collection, model training, and evaluation, using both teleoperation data and autonomously collected data. Specifically, teleoperation data in simulation was gathered using LeapMotion, while an exoskeleton device was employed for real-world teleoperation data collection. The performance of policies trained on teleoperation data was then compared with that of policies trained on autonomously collected data, with the results summarized in Tab. 6.

Experimental results indicate that policies trained on teleoperated and autonomously collected data achieve comparable performance during evaluation. However, from the perspective of human

Table 6: Performance Comparison of HALO with **Different Data Sources**

| Method (Data Source) | Hang Tops (Simulation) | Wear Bowlhat (Simulation) | Fold Tops (Real-World) |
|---|---|---|---|
| HALO (Automated Collection Data) | 0.92±0.04 | 0.72±0.04 | 13 / 15 |
| HALO (Teleoperation Data) | 0.88±0.03 | 0.70±0.06 | 13 / 15 |

and time cost, teleoperation-based data collection is significantly more labor-intensive and time-consuming than automated data collection. It is also important to note that our data is not synthetically generated, but collected through real executions in either simulation or the real world, which means the demonstrations are physically realistic. The entire data collection pipeline is enabled by model inference and one-shot demonstrations.

# G   Additional Baseline Comparison

We have additionally included four new baselines for comparison: **ACT (IL), pi0 (VLA), RDT (VLA), and Eureka (RL+VLM)**. We selected representative tasks for evaluation, including **Fling Dress, Fold Trousers, Hang Coat, and Wear Bowlhat in simulation, as well as Fold Tops in real world**. The evaluation protocol remains consistent with the original paper:

- For simulation tasks, each task is evaluated over 50 episodes using three different random seeds. We report the success rates as Mean ± Std across all trials.

- For real-world task, we evaluate using 3 distinct garments per category, each tested under 5 different initial deformations. We report the success rates as successful_trials / all_trials.

Table 7: Performance Comparison of **More Baseline Methods** on Typical Tasks

| Method | Fling Dress (Simulation) | Fold Trousers (Simulation) | Hang Coat (Simulation) | Wear Bowlhat (Simulation) | Fold Tops (Real World) |
|---|---|---|---|---|---|
| DP (IL) | 0.59±0.05 | 0.47±0.04 | 0.52±0.04 | 0.41±0.05 | *9/15* |
| DP3 (IL) | 0.51±0.03 | 0.54±0.07 | 0.58±0.04 | 0.55±0.04 | 8/15 |
| ACT (IL) | 0.35±0.02 | 0.49±0.06 | 0.43±0.04 | 0.51±0.03 | 7/15 |
| pi0 (VLA) | *0.69±0.01* | 0.52±0.06 | *0.72±0.01* | *0.59±0.02* | 10/15 |
| RDT (VLA) | 0.60±0.02 | *0.58±0.02* | 0.62±0.02 | 0.48±0.01 | 9/15 |
| Eureka (RL+VLM) | / | / | 0.16±0.03 | 0.08±0.02 | / |
| HALO (Ours) | **0.82±0.06** | **0.77±0.02** | **0.90±0.01** | **0.72±0.04** | **13/15** |

Analysis For **Imitation-Learning-Based Methods**:

We collect 100 demonstrations for each simulation task and collect 50 demonstrations for each real-world task, which are used for policy training (DP/DP3/ACT). A notable limitation of their performance lies in the inability to accurately grasp the target region of the garment, as well as the lack of fine-grained control during placement based on the garment's shape(/state).

Analysis for **VLA Methods**:

We combine all the demonstrations of simulation and real-world tasks to fine-tune pi0 and RDT model. In terms of final performance, pretrained models such as pi0 and RDT outperform from-scratch approaches like DP, DP3, and ACT. However, compared to HALO, VLA-based models still exhibit clear limitations in accurately perceiving garment shape and state, and in executing precise grasp-and-place actions, resulting in a noticeable performance gap relative to HALO.

Analysis for **RL-Based Methods**:

We selected Eureka as the representative RL-VLM-combined method and conducted preliminary experiments on several tasks in simulation. While we observed some initial successful cases on simpler tasks such as Hang Coat and Wear Bowlhat, the overall success rate remained low. Additionally, due to the lack of robust parallelization support in the current Isaac Sim environment and the limited effectiveness of reward functions in handling long-horizon tasks like Fling and Fold, this baseline did not yield valid results across all tasks. Nevertheless, we are actively developing a multi-parallel deformable object manipulation environment based on Isaac Lab, and RL-based approaches for deformable manipulation will be a key focus of our future work.

# H  Training Details of Main Algorithm

## H.1  Garment Affordance Model (GAM)

**Hyper-Parameters Selection**. The training of GAM follows the hyperparameter settings used in the UniGarmentManip framework. we set the number of skeleton pairs to be 50 and batch size to be 32. In each batch, we sample 32 garment pairs. For each garment pair, we sample 20 positive and 150 negative point pairs for each positive point pair. Therefore, in each batch, $32 \times 32 \times 20$ data will be used to update the model. During the Correspondence training stage, we train the model for 40,000 batches. 'Coarse-to-fine Refinement' and 'Few-shot Adaptation' mentioned in UniGarmentManip are also adopted for improving GAM's performance.

**Computational Resource**. we use PyTorch as our Deep Learning framework. Each experiment is conducted on an RTX 4090 GPU, and consumes about 22 GB GPU Memory for training. It takes about 20 hours to train the Coarse Stage, with 1-2 hours of Coarse-to-Fine Refinement and 0.5 hour's Few-Shot Adaptation.

We trained GAM checkpoints for eight types of garments, including: **Baseball_Cap**, **Dress_LongSleeve**, **Glove**, **Scarf**, **Tops_FrontOpen**, **Tops_LongSleeve**, **Tops_NoSleeve**, and **Trousers**. We release all these checkpoints in our code repo.

## H.2  Structure-Aware Diffusion Policy (SADP)

**Hyper-Parameters Selection.** We set the *horizon*, *observation_steps*, and *action_steps* to 8, 3, and 4, respectively. The number of denoising steps in the diffusion process is set to 10. The model is trained for a total of 3000 epochs, with validation performed every 25 epochs and checkpoints saved every 100 epochs. We use the AdamW optimizer with an initial learning rate of $1 \times 10^{-4}$, and adopt a cosine learning rate scheduler with 500 warm-up steps. During dataset loading, the test set is configured to comprise 2% of the entire training dataset.

**Computational Resource**. We use PyTorch as our deep learning framework. All experiments are conducted on an NVIDIA A800 GPU, with approximately 75 GB of GPU memory consumption when training with a batch size of 200. The training process takes around 16 hours to complete 3000 epochs. However, checkpoints from earlier epochs can be selected for validation if desired.

We release two variants of our method: **SADP** and **SADP_G**. **SADP** is designed for *Garment-Environment-Interaction Tasks*, where the encoder incorporates the point cloud of environment-interaction objects. In contrast, **SADP_G** is tailored for *Garment-Self-Interaction Tasks*, where the encoder excludes the point cloud of environment-interaction objects. We provide detailed tutorials for both SADP and SADP_G in the released codebase to facilitate easy usage and integration.

# I  Limitation

The manipulation of deformable objects (garments) is a highly challenging field. ***DexGarmentLab*** provides an initial environment, along with data collection and policy training methods, to facilitate advancements in this domain. However, we must acknowledge that DexGarmentLab still has several limitations, which will be analyzed in detail as follows.

## I.1  Garment Simulation Method Limitation

In DexGarmentLab, We employ Position-Based-Dynamics (PBD) and Finite-Element-Method (FEM) to simulate garment. However, both PBD and FEM have different limitations to accurately simulate the real-performance of fabric.

### I.1.1  Position-Based-Dynamics (PBD)

In the simulator, garments modeled using Position-Based Dynamics (PBD) are represented as a collection of discrete particles, as shown in Fig. 13. This approach effectively captures the softness of fabric, facilitating operations such as folding and deformation. For this reason, we employ PBD to simulate larger and highly deformable garments, such as tops, trousers, dresses, etc.

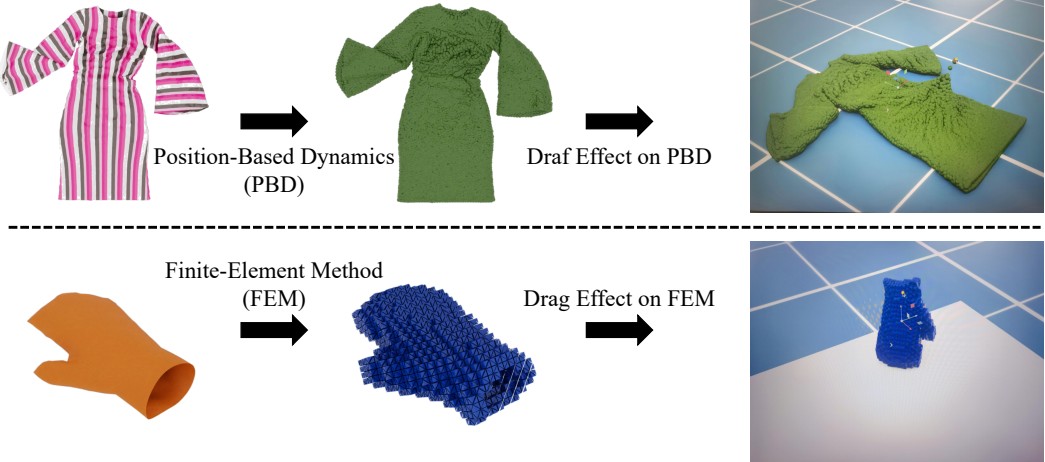

Figure 13: **Garment Simulation Method**

However, since the particles are not tightly connected and contain numerous gaps (shown in Fig. 13), and self-collision interactions between particles can easily lead to instability in their states (This instability manifests as continuous jittering of the garment.), when manipulating the garment, penetration artifacts may occur, and maintaining a stable configuration can be challenging.

In DexGarmentLab, we designed a set of carefully tuned parameters for PBD simulation tailored to the provided assets, aiming to mitigate these issues. While some unrealistic garment behaviors may still persist, our approach enables a relatively reasonable approximation of garment performance.

### I.1.2 Finite-Element-Method (FEM)

In the simulator, garments modeled using Finite Element Method (FEM) are represented as a collection of discrete blocks, as shown in Figure 13. This approach effectively simulates non-deformable and elastic objects but does not accurately capture garment deformations. When garments undergo deformation in the simulator, they tend to revert to their original shape easily, a phenomenon that is particularly pronounced for larger garments. As shown in Fig. 13, when garments modeled using FEM are dragged, the discrete blocks used for simulation remain tightly connected and do not separate. Besides, the garment tends to preserve its original shape throughout the dragging process.

In DexGarmentLab, we employ FEM to simulate garments such as gloves and hats, which do not exhibit significant deformations in the real world and typically maintain relatively stable shapes. Within the corresponding tasks, these garments demonstrate highly realistic physical properties.

### I.2 Environment Task Limitation

Although we propose 15 different tasks, including garment-self-interaction tasks and garment-environment-interaction tasks, these tasks all belong to single-garment tasks, while multi-garments tasks are not introduced.

What's more, in our simulation environment, we utilize two independent robotic arms equipped with dexterous hands (UR10e + ShadowHand). The tasks in our scenarios do not involve the movement of robotic system. However, for real-world household applications, a more suitable robotic platform would be a dual-arm mobile robot with a wheeled base. Nevertheless, our proposed method is applicable to various types of dual-arm robotic systems.

The aforementioned environment task limitations are fundamentally challenging issues in the field of deformable object manipulation and even in robotics as a whole. We look forward to future researchers building upon DexGarmentLab to continuously address these challenges and advance the development of deformable object manipulation.

## I.3 Generalizable Policy Limitation

Based on both simulation and real-world experimental results, we summarize the following points regarding policy limitation:

1. For garments with occluded or highly deformed regions, the Garment Affordance Model (GAM) does not always predict manipulation points accurately. This can lead to failed or suboptimal grasps by the dexterous hand, impacting the final performance. Nonetheless, GAM performs reliably on most garments. For challenging cases, an initial unfold action is recommended to expose key regions.

2. The policy struggles with garments that deviate from standard geometries, such as asymmetrical designs (e.g., single-sleeved tops) or heavily adorned costumes, which distort the point cloud and affect both GAM and SADP performance.

## J  DexGarmentLab Assets

- **Garment Assets.** We select garments from ClothesNet [45], a large-scale dataset of 3D clothes objects with information-rich annotations. We select garments from 8 categories (including Tops, Dress, Trousers, Hat, Scarf, etc.) and use two physical simulation method (FEM and PBD) to simulate them.

- **Robots Assets.** The dual-arm robot used in DexGarmentLab consists of two ShadowHands mounted on UR10e robotic arms. Leveraging the URDF files provided by [11], we integrated the ShadowHand and UR10e URDFs and converted the combined URDF into a USD file using NVIDIA Isaac Sim. Users can customize the robot configuration through the provided URDF file to suit their specific tasks or simulation setups.

- **Environment-Interaction Assets.** The environment-interaction assets used in DexGarment-Lab mainly includes: hanger, pothook, placement platform, human. The hanger, pothook and placement platform are created using basic components (for example, cube, capsule, etc.) supplied by Isaac Sim, while human model are obtained from *Omniverse Base Asset library*.

- **Material Assets.** Materials are crucial components of virtual relightable assets, defining the interaction of light at the surface of geometries. Our materials are primarily sourced from two repositories: a selection from *Omniverse Base Material Library* and additional assets obtained from *https://ambientcg.com/*. These materials are mainly used for simulating garments, grounds, and other scene elements.

## K  Detailed Task Description

As we mentioned in the paper, we divide all the tasks into two categories: Garment-Self-Interaction Task and Garment-Environment-Interaction Task.

Garment-Self-Interaction Tasks include:

**Fold Tops** (Sec. K.1), **Fold Dress** (Sec. K.2), **Fold Trousers** (Sec. K.3), **Fling Tops** (Sec. K.4), **Fling Dress** (Sec. K.5), **Fling Trousers** (Sec. K.6).

Garment-Environment-Interaction Tasks include:

**Hang Coat** (Sec. K.7), **Hang Tops** (Sec. K.8), **Hang Dress** (Sec. K.9), **Hang Trousers** (Sec. K.10), **Store Tops** (Sec. K.11), **Wear Baseball Cap** (Sec. K.12), **Wear Bowl Hat** (Sec. K.13), **Wear Scarf** (Sec. K.14), **Wear Glove** (Sec. K.15).

In this section, we will provide a detailed description of each selected task, covering *task initialization and randomization*, *task sequences*, *task success metrics*, *garment assets for task* and other related aspects. These details will be introduced in separate subsections for clarity.

### K.1  Fold Tops

#### K.1.1  Task Initialization and Randomization

Task Configuration is shown in Fig. 14 (a).

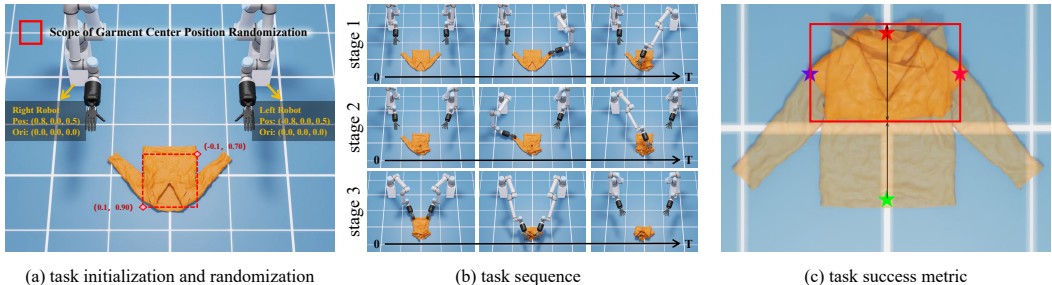

| (a) task initialization and randomization | (b) task sequence | (c) task success metric |

Figure 14: **Fold Tops Task.**

As for initialization, the garment is initialized in a flat state, which means that the initial orientation of the garment is set to (0.0, 0.0, 0.0). The positions of the left and right robots are set to (-0.8, 0.0, 0.5) and (0.8, 0.0, 0.5) respectively, with both orientations initialized to (0.0, 0.0, 0.0).

During data collection and policy evaluation, we randomly selected the position of the garment within a certain range. For this task, the initial position of the tops is randomized within a rectangular area defined by -0.10<x<0.10 and 0.70<y<0.90.

It should be noted that the randomization range for the position refers to the location of garment's center, and the unit distance is 1 meter. Additionally, the initial orientation is set as euler angles. All descriptions in the following sections follow this convention.

### K.1.2 Task Sequence

As shown in Fig. 14 (b), the sequence of *Fold Tops* consists of three stages. First, fold the left sleeve to the right, then fold the right sleeve to the left, and finally grab the corners of the garment and fold them upward to complete the fold.

### K.1.3 Task Success Metrics

The garment is initially in a flat state. First, Garment Affordance Model (GAM) is used to locate four key points: the left and right sleeve points, as well as the collar and bottom of the garment. Then, the left and right boundaries are determined by the points on the left and right sleeves, and the central boundary is determined by the points on the collar and bottom. The collar and center are then used as the upper and lower boundaries, respectively, forming a rectangular area, as shown in Fig. 14 (c). The success of the fold is determined by checking whether the final state lies within this area. If the garment region within the area covers more than 85% of the total garment area, the 'fold' manipulation is considered successful. To ensure data validity, images of the final state were also recorded and manually checked.

### K.1.4 Garment Assets for Task

For this task, the garments used are long-sleeved tops, a total of **247** pieces. During data collection, we first selected 100 garments and then randomly chose from these 100 garments for data collection. During policy randomization, we randomly select from all 247 garments to ensure that the validation set contains data that was not seen during training.

## K.2 Fold Dress

### K.2.1 Task Initialization and Randomization

Task Configuration is shown in Fig. 15 (a).

The initialization configuration of Fold Dress task is the same with those of Fold Tops. As for randomization, the initial position of the dress is randomized within a rectangular area defined by -0.10<x<0.10 and 0.65<y<0.90.

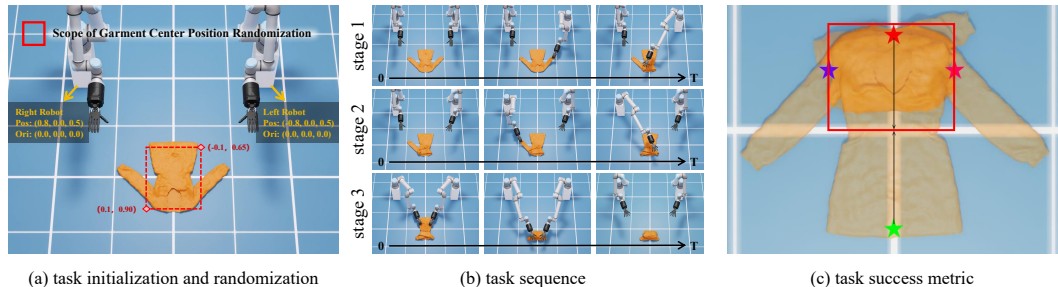

| | (a) task initialization and randomization | (b) task sequence | (c) task success metric |

Figure 15: **Fold Dress Task.**

### K.2.2 Task Sequence

As shown in Fig. 15 (b), the sequence of *Fold Dress* also consists of three stages. First, fold the left sleeve to the right, then fold the right sleeve to the left, and finally, since a dress is usually longer, we choose to grab the waist of the dress (instead of the skirt hem) and fold it upward.

### K.2.3 Task Success Metrics

The success criteria for this task are the same as for Tops. Four points are selected to form a rectangular area, as shown in Fig. 15 (c), and the folded garment is checked to see if it lies within this area. If the garment region within the area covers more than 80% of the total garment area, the 'fold' manipulation is considered successful.

### K.2.4 Garment Assets for Task

The garments used in this task are long-sleeved dresses, with a total of **38** pieces. During data collection, we first randomly select 18 dresses and then randomly choose from these 26 for data collection. In the validation experiments, dresses are randomly selected from all 38 pieces for the validation process.

## K.3 Fold Trousers

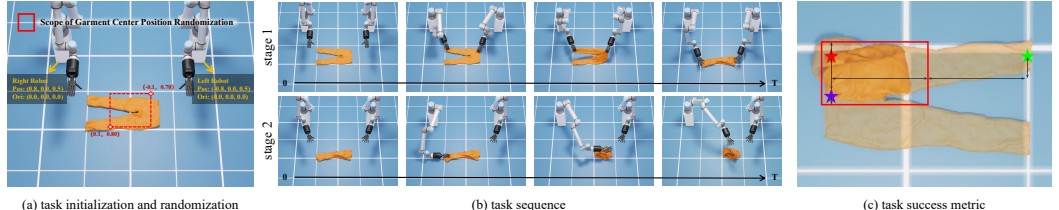

| (a) task initialization and randomization | (b) task sequence | (c) task success metric |

Figure 16: **Fold Trousers Task.**

### K.3.1 Task Initialization and Randomization

Task Configuration is shown in Fig. 16 (a).

For the initialization of Fold Trousers, the trousers need to be laid horizontally on the ground, so the orientation of the garment is set to (0.0, 0.0, 90.0). As for randomization, the initial position of the trousers is randomized within a rectangular area defined by -0.10<x<0.10 and 0.70<y<0.90.

### K.3.2 Task Sequence

The sequence of *Fold Trousers* consists of two stages, as shown in Fig. 16 (b). First, fold the trousers along the center axis, then fold the pant legs towards the waistband to complete the task.

### K.3.3 Task Success Metrics

Similar to the previous two folding tasks, three points are selected, as shown in Fig. 16 (c). The boundaries of the folded garment are calculated based on the positions of these three points, and

the success of the task is determined by checking whether the folded garment lies within this area. If the garment region within the area covers more than 85% of the total garment area, the 'fold' manipulation is considered successful.

### K.3.4 Garment Assets for Task

The Fold Trousers task uses a total of **317** trousers. During data collection, 100 pieces are randomly selected from the 317, and then randomly chosen from these 100 for data collection. In validation experiments, garments are randomly selected from all 317 pieces for validation.

## K.4 Fling Tops

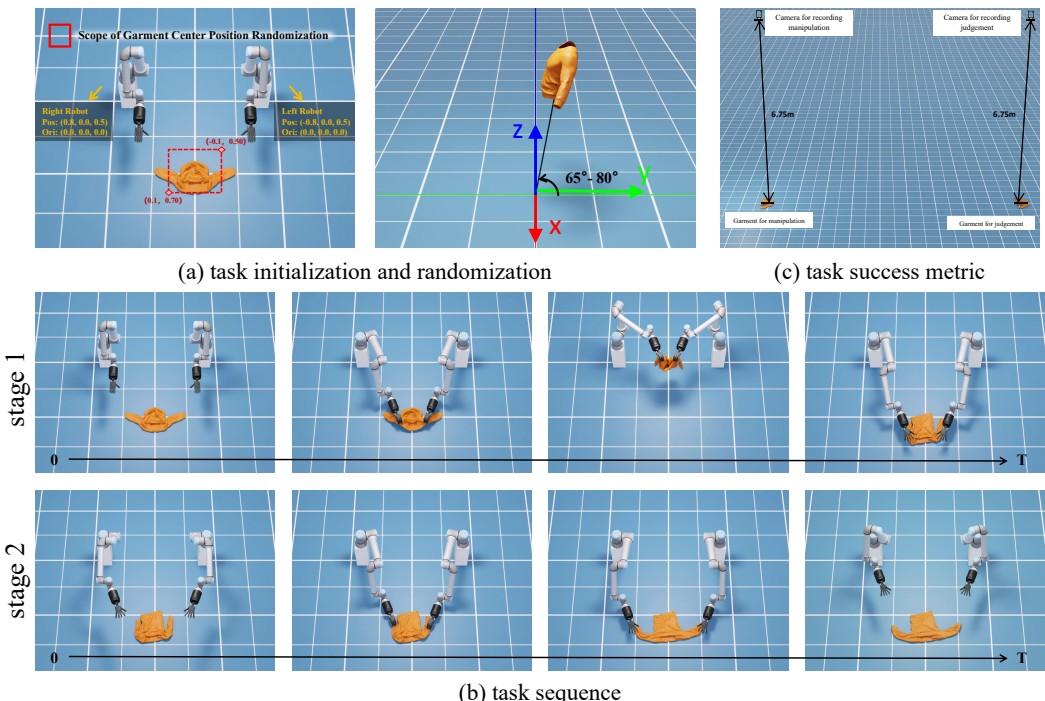

(a) task initialization and randomization      (c) task success metric

(b) task sequence

Figure 17: **Fling Tops Task.**

### K.4.1 Task Initialization and Randomization

Task Configuration is shown in Fig. 17 (a).

For the Fling task, the garment should initially be in a wrinkled state. Therefore, an inclination angle and a certain height are given at the start to ensure that the garment lands in a stacked state. For this task, the inclination angle is set to 65-80 degrees, i.e., the initial orientation is (65.0-80.0, 0.0, 0.0), as shown in Fig. 17 (a).

As for position randomization, the initial position of the tops is randomized within a rectangular area defined by -0.10<x<0.10 and 0.50<y<0.70.

### K.4.2 Task Sequence

As shown in Fig. 17 (b), the *Fling Tops* task consists of two stages. First, grab the shoulders of the top, lift it up, and extend hands forward and downward to flatten the body part of the top. Then, grab both sleeves and pull them outward to flatten the sleeves.

### K.4.3 Task Success Metrics

For the Fling task, we use an area-based judgment method. A matching garment is placed in a different location in the environment in a flat, non-tilted state. In the final judgment, images are

captured by cameras positioned at the same height directly above both the Fling garment and the garment used for comparison, as shown in Fig. 17 (c). Then, the proportion of the area occupied by the garments in the images is calculated. If the difference in area proportions is smaller than a certain threshold (here we set 0.2), it indicates that the area of the Fling garment is close to that of the flat garment, and the Fling is considered successful.

### K.4.4 Garment Assets for Task

The assets and partitioning method used in this task are the same as those in the Fold Tops task mentioned earlier.

### K.5 Fling Dress

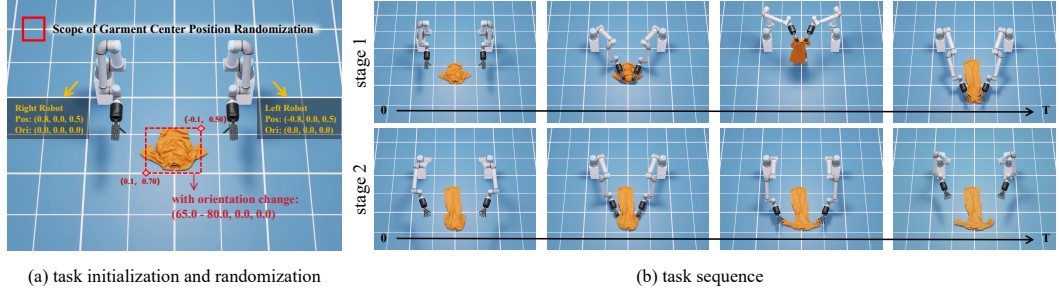

(a) task initialization and randomization        (b) task sequence

Figure 18: **Fling Dress Task.**

### K.5.1 Task Initialization and Randomization

Task Configuration is shown in Fig. 18 (a).

In this task, the orientation randomization of the dress is also set to (65.0-80.0, 0.0, 0.0). As for position randomization, the initial position of the dress is randomized within the range -0.10<x<0.10 and 0.50<y<0.70.

### K.5.2 Task Sequence

The sequence of *Fling Dress* is essentially the same as that of Fling Tops, as shown in Fig. 18 (b).

### K.5.3 Task Success Metrics

The same area-based judgment method as described in the Fling Tops task is also applied in this task.

### K.5.4 Garment Assets for Task

The assets and partitioning method used in this task are the same as those in the Fold Dress task mentioned earlier.

### K.6 Fling Trousers

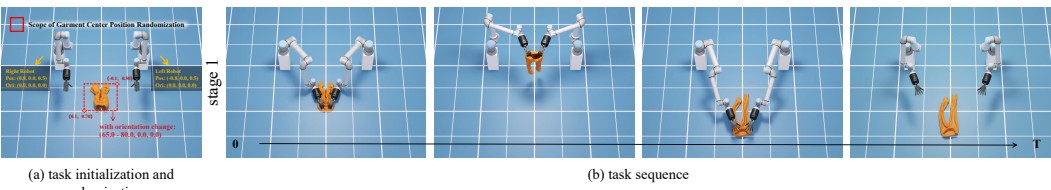

(a) task initialization and randomization        (b) task sequence

Figure 19: **Fling Trousers Task.**

### K.6.1 Task Initialization and Randomization

Task Configuration is shown in Fig. 19 (a).

In this task, the orientation randomization of the trousers is also set to (65.0-80.0, 0.0, 0.0). As for position randomization, the initial position of the trousers is randomized within the range -0.10<x<0.10 and 0.50<y<0.70.

### K.6.2 Task Sequence

The *Fling Trousers* task consists of only one stage: grab the waistband and lift it up, then extend hands forward and downward to flatten the trousers, as shown in Fig. 19 (b).

### K.6.3 Task Success Metrics

The assets and partitioning method used in this task are the same as those in the Fold Trousers task mentioned earlier.

### K.6.4 Garment Assets for Task

The same area-based judgment method as described in the Fling Tops task is also applied in this task.

## K.7 Hang Coat

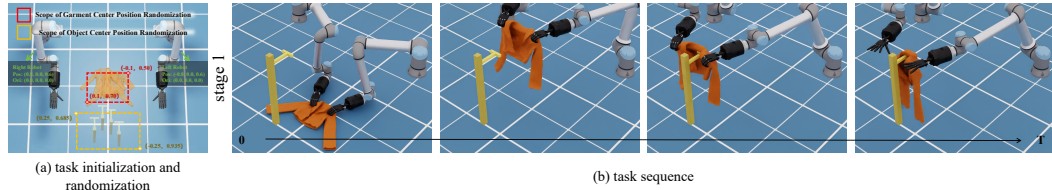

(a) task initialization and randomization

(b) task sequence

Figure 20: **Hang Coat Task.**

### K.7.1 Task Initialization and Randomization

Task Configuration is shown in Fig. 20 (a).

This task involves randomization not only in the position of garments but also in the position of environment-interaction objects. The initial position of the coats is randomized within the range -0.10<x<0.10 and 0.50<y<0.70. The initial position of the pothook is randomized within the range -0.25<x<0.25 and 0.685<y<0.935.

### K.7.2 Task Sequence

The *Hang Coat* task consists of one stage: grab the coat's lapels and lift the garment up to the pothook, as shown in Fig. 19 (b). The lifting height and placement position vary depending on the shape of the garment and the location of the pothook.

### K.7.3 Task Success Metrics

The ideal final state of this task is that the garment is stably hung on the pothook without falling off. Therefore, we determine task success based on the center position of the garment. Specifically, if the z-coordinate of the garment's center is greater than 0.5 (to exclude cases where it has fallen to the ground) and less than 2.0 (to exclude abnormal simulation states), the task is considered successful.

### K.7.4 Garment Assets for Task

The garments used in the Open Coats task are front opening tops, with a total of **78** garments. During data collection, 40 pieces of the garments are randomly selected, and then a random subset of these 40 garments is chosen for data collection. In the validation experiments, garments are randomly selected from all 78 pieces for validation.

## K.8 Hang Tops

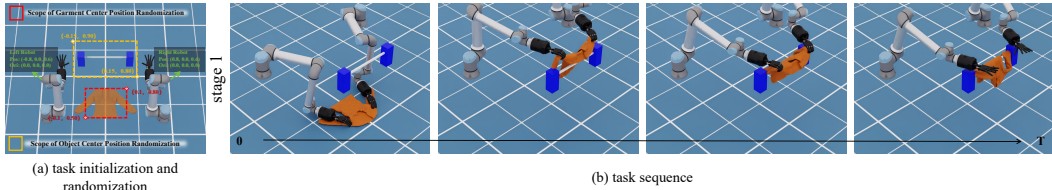

(a) task initialization and randomization

(b) task sequence

Figure 21: **Hang Tops Task.**

### K.8.1 Task Initialization and Randomization

Task Configuration is shown in Fig. 21 (a).

This task involves randomization not only in the position of garments but also in the position of environment-interaction objects. The initial position of the tops is randomized within the range -0.10<x<0.10 and 0.50<y<0.80. The initial position of the hanger is randomized within the range -0.15<x<0.15 and 0.80<y<0.90.

### K.8.2 Task Sequence

The sequence of *Hang Tops* is shown in Figure 21 (b). First, grab the shoulders of the top and lift the garment, then move it forward to drape the top over the hanger. The lifting height and placement position vary depending on the shape of the garment and the location of the hanger.

### K.8.3 Task Success Metrics

For the Hang task, the garment should ultimately be hanging on the hanger, with all parts of the garment above a certain height. Therefore, we determine the success of the Hang task by checking whether all points of the garment's point cloud are above this height. Additionally, we also check whether the distance between the center of the garment and the center of the hanger falls within a certain range, in order to exclude cases where the garment is not hung near the central region of the rack.

### K.8.4 Garment Assets for Task

The assets and partitioning method used in this task are the same as those in the Fold Tops task mentioned earlier.

## K.9 Hang Dress

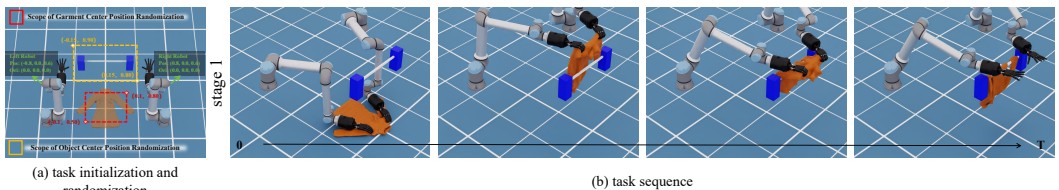

(a) task initialization and randomization

(b) task sequence

Figure 22: **Hang Dress Task.**

### K.9.1 Task Initialization and Randomization

Task Configuration is shown in Fig. 22 (a).

This task involves randomization not only in the position of garments but also in the position of environment-interaction objects. The initial position of the dress is randomized within the range -0.10<x<0.10 and 0.50<y<0.80. The initial position of the hanger is randomized within the range -0.15<x<0.15 and 0.80<y<0.90.

### K.9.2 Task Sequence

The sequence of *Hang Dress* is the same as that of Hang Tops, as shown in Figure 22 (b). The lifting height and placement position vary depending on the shape of the garment and the location of the hanger.

### K.9.3 Task Success Metrics

The metric for Hang Dress is the same as that for Hang Tops.

### K.9.4 Garment Assets for Task

The assets and partitioning method used in this task are the same as those in the Fold Dress task mentioned earlier.

## K.10 Hang Trousers

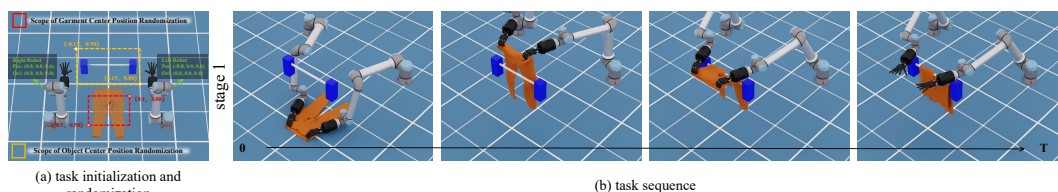

(a) task initialization and randomization

(b) task sequence

Figure 23: **Hang Trousers Task.**

### K.10.1 Task Initialization and Randomization

Task Configuration is shown in Fig. 23 (a).

This task involves randomization not only in the position of garments but also in the position of environment-interaction objects. The initial position of the dress is randomized within the range -0.10<x<0.10 and 0.50<y<0.80. The initial position of the hanger is randomized within the range -0.15<x<0.15 and 0.80<y<0.90.

### K.10.2 Task Sequence

The process of Hang Trousers is shown in Fig. 23 (b). Grab the waistband and then follow the same steps as for Hang Tops and Dress. The lifting height and placement position vary depending on the shape of the garment and the location of the hanger.

### K.10.3 Task Success Metrics

The metric for Hang Trousers is the same as that for Hang Tops.

### K.10.4 Garment Assets for Task

The assets and partitioning method used in this task are the same as those in the Fold Trousers task mentioned earlier.

## K.11 Store Tops

### K.11.1 Task Initialization and Randomization

Task Configuration is shown in Fig. 24 (a).

This task involves randomization not only in the position of garments but also in the position of environment-interaction objects. The initial position of the tops is randomized within the range -0.05<x<0.05 and 0.65<y<0.75. The initial position of the placement platform is randomized within the range -0.30<x<0.30 and 1.00<y<1.20.

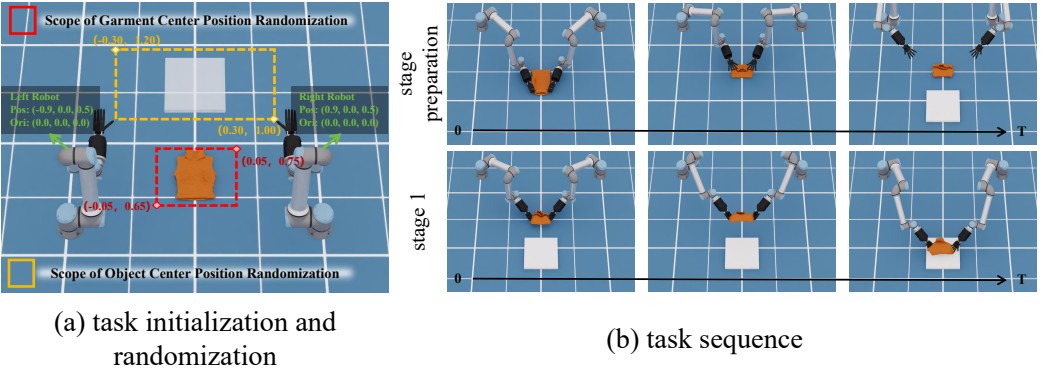

(a) task initialization and randomization

(b) task sequence

Figure 24: **Store Tops Task.**

### K.11.2 Task Sequence

The sequence of *Store Tops* is divided into stage_preparation and stage_1, as shown in Figure 24 (b). In stage_preparation, grab the shoulders and fold the garment towards the corners in order to make garment ready. Then in stage 1, use both hands to grab the corners and place the garment in the corner of placement platform.

### K.11.3 Task Success Metrics

The goal of this task is to place the folded garment at the exact center of the placement platform. Therefore, after the task is completed, we load a camera above the garment to capture a point cloud of its final state. The center position of the garment is then computed from the point cloud coordinates and compared with the center position of the placement platform, which is accessible in the simulator. If the distance between the two centers is smaller than a predefined threshold (set to 0.1 in this case), the task is considered successful.

### K.11.4 Garment Assets for Task

The garments used for the Store task are sleeveless tops, with a total of **217** pieces. During data collection, 100 pieces are randomly selected from these 217 garments. In validation experiments, garments are randomly selected from all 217 pieces.

### K.12 Wear Baseball Cap

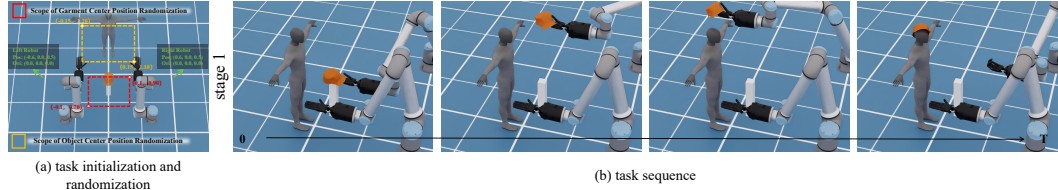

(a) task initialization and randomization

(b) task sequence

Figure 25: **Wear Baseball Cap Task.**

### K.12.1 Task Initialization and Randomization

Task Configuration is shown in Fig. 25 (a).

This task involves randomization not only in the position of garments but also in the position of environment-interaction objects. The initial position of the baseball cap is randomized within the range -0.10<x<0.10 and 0.70<y<0.90. The initial position of the human is randomized within the range -0.15<x<0.15 and 1.10<y<1.20.

### K.12.2 Task Sequence

The sequence of *Wear Baseball Cap* is shown in Figure 25 (b). First, use the right hand to grab the brim of the cap, then position the cap and place it on the top of the head. The lifting height and placement position vary depending on the shape of the garment and the location of the human.

### K.12.3 Task Success Metrics

The final goal of wearing the cap is for it to be placed on top of the head. Therefore, the success criterion is based on whether the distance between the final center of the cap and the top of the head is within a certain threshold.

### K.12.4 Garment Assets for Task

There are **12** baseball caps in total. During data collection and policy validation, we all use 12 pieces of caps for random selection.

## K.13 Wear Bowl Hat

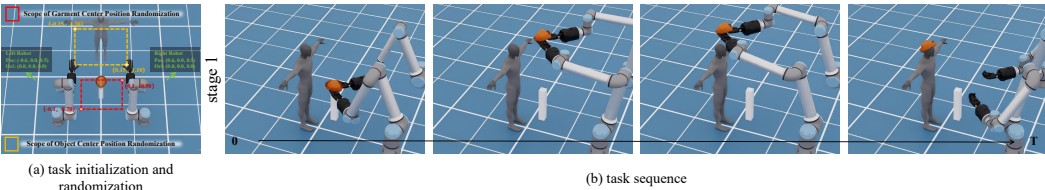

(a) task initialization and randomization

(b) task sequence

Figure 26: **Wear Bowl Hat Task.**

### K.13.1 Task Initialization and Randomization

Task Configuration is shown in Fig. 26 (a).

This task involves randomization not only in the position of garments but also in the position of environment-interaction objects. The initial position of the bowl hat is randomized within the range -0.10<x<0.10 and 0.70<y<0.90. The initial position of the human is randomized within the range -0.15<x<0.15 and 1.10<y<1.20.

### K.13.2 Task Sequence

The sequence of *Wear Bowl Hat* is done with both hands. First, use both hands to grab the left and right sides of the hat from below, then align it with the head, and gently place it on the head. The entire process is shown in Figure 26 (b). The lifting height and placement position vary depending on the shape of the garment and the location of the human.

### K.13.3 Task Success Metrics

The same as the criterion for wearing baseball cap.

### K.13.4 Garment Assets for Task

There are **8** bowl hats in total. During data collection and policy validation, we all use 8 pieces of hat for random selection.

## K.14 Wear Scarf

### K.14.1 Task Initialization and Randomization

Task Configuration is shown in Fig. 27 (a).

This task involves randomization not only in the position of garments but also in the position of environment-interaction objects. The initial position of the scarf is randomized within the range

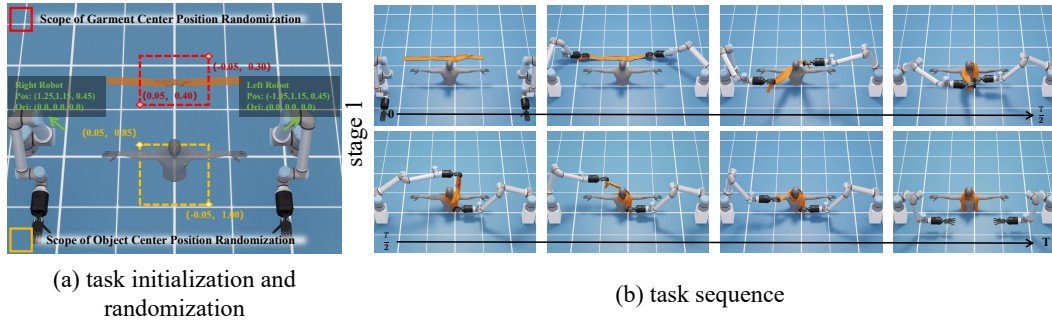

<table>
<tr><td>(a) task initialization and randomization</td><td>(b) task sequence</td></tr>
</table>

Figure 27: **Wear Scarf Task.**

-0.05<x<0.05 and 0.30<y<0.40. The initial position of the human is randomized within the range -0.05<x<0.05 and 0.85<y<1.00.

### K.14.2   Task Sequence

The sequence of *Wear Scarf* is shown in Figure 27 (b). First, grab both ends of the scarf and drape it around the neck. Then, use the right hand to grab the right end and wrap it around the neck to complete the task. The manipulation position and placement position vary depending on the shape of the garment and the location of the human.

### K.14.3   Task Success Metrics

For the scarf-wearing task, if the scarf is not successfully placed around the person's neck, it will droop to the ground. To detect this, we place a camera in front of and behind the human model to capture point clouds of the scarf. If the number of points below a certain height threshold (set to 0.2 meters) exceeds a predefined count threshold (set to 20) in both front and rear point clouds, we consider the scarf to be drooping on the ground and thus regard the task as a failure. Otherwise, the task is considered successful.

### K.14.4   Garment Assets for Task

We use 8 types of scarf for data collection and randomization, which have different length (0.35m, 0.36m, 0.37m, 0.38m, 0.39m, 0.40m, 0.41m, 0.42m, respectively).

### K.15   Wear Glove

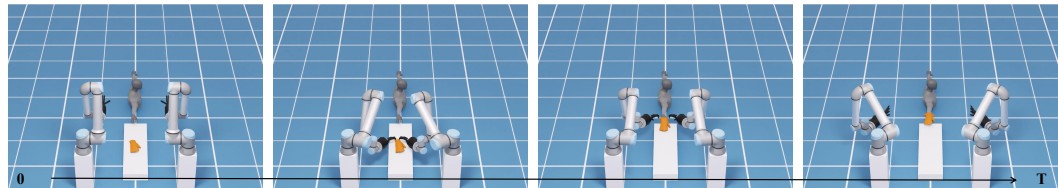

Figure 28: **Wear Glove Task.**

The sequence of the *Wear Gloves* task is shown in Figure 28. First, use the index fingers of both hands to insert into the gloves and pull them apart. Then, move the gloves forward and put them on the hands.

Due to the small size of the glove and the requirement for precise insertion of the index finger, followed by pinching with the thumb to grasp the side of the glove, achieving this effect is challenging with the current FEM simulation and dexterous hand manipulation. Therefore, we employed the *attachment block* method to accomplish the wear gloves task, and did not implement randomization of the initial position for this task, nor did we validate it through policy evaluation. We look forward to future work enabling dexterous hands to better handle such extremely fine-grained tasks.

### K.16   Additional Explanation for Success Metric

Although we have designed specific success metrics for each task, and our validation shows that these metrics can reliably determine task success or failure in most cases, the inherently complex nature of garment states makes it difficult to judge success purely based on the final state in certain situations. Therefore, during both data collection and policy validation, we record the final state of the garment as well as a full video of each episode. During data collection, users can review the final state images to identify potentially abnormal episodes and use the corresponding videos to decide whether to keep the data. Similarly, during policy validation, users can apply the same approach to avoid misjudgments in result evaluation.

## L   Broader Impact

Our work presents a dexterous garment manipulation environment, with a particular focus on coordinated dual-arm dexterous hands. This effort advances the field of deformable object manipulation and lays a solid foundation—both in simulation and algorithmic development—for future progress in general-purpose home-assistant robotics involving deformable object handling. We haven't observed negative potential impacts.

