# OpenReview forum: "DexGarmentLab: Dexterous Garment Manipulation Environment with Generalizable Policy"
_NeurIPS.cc/2025/Conference — NeurIPS 2025 spotlight_

### Official Review · Reviewer_ntYt · 2025-06-19

**Clarity:** 3
**Significance:** 3
**Originality:** 2
**Rating:** 5
**Confidence:** 4

**Summary:**

The paper introduces DexGarmentLab, a comprehensive framework for bimanual dexterous garment manipulation. Contributions include: (1) A new simulation environment built on Isaac Sim, featuring over 2,500 garment assets from ClothesNet across 15 task scenarios; (2) an automated data collection pipeline that is able to generate datasets of diverse trajectories from a single expert demonstration using a Garment Affordance Model (GAM) to identify manipulation points on new garments; (3) a hierarchical policy HALO which uses GAM to generate manipulation trajectories conditioned on visual and state information. The approach is validated in both simulation and in the real world.

**Questions:**

- The entire data collection process is bootstrapped from a single, manually defined expert demonstration. How does the system perform if this initial demonstration is varied? For instance, if you provide a slightly different (but still successful) folding sequence, how does that affect the diversity and quality of the 100 generated demonstrations and the performance of the resulting policy?
- Could the authors comment on the extensibility of DexGarmentLab to facilitate future research? For example, what is the workflow for tuning the simulation parameters if a new garment is introduced? The paper mentions parameters were "carefully tuned"; is there a systematic procedure for this, or does each new asset require extensive, ad-hoc manual tuning? How readily can the policy and data collection frameworks be adapted to different hardware? How hard would it be to map the 60-DoF action space and the LeapMotion data collection to a new bimanual setup with different kinematics?

**Ethical Concerns:**

["NO or VERY MINOR ethics concerns only"]

**Final Justification:**

During the rebuttal, all my questions and concerns about the other two core components of the manuscript were properly addressed by the authors. I believe this work is a nice addition to the dexterous robotics research community, with contributions spanning from infrastructure to algorithms. Therefore, I give a final recommended score of Accept.

**Limitations:**

Yes, the authors have done a thorough discussion of the limitations in Appendix F.

**Paper Formatting Concerns:**

See weaknesses.

**Quality:**

3

**Strengths And Weaknesses:**

Strengths:
- [Quality]: The DexGamentLab ecosystem is a complete package with a feature-rich simulation environment, a large-scale asset library, an automated data collection methodology, and a policy. 14 simulation tasks and 4 real robot tasks further strengthen its statistical significance.
- [Clarity]: The paper is generally well-organized and easy to follow, considering the amount of work done.
- [Significance]: The framework provides the first bimanual dexterous garment manipulation environment at this scale. Automated data collection reduces reliance on teleoperation. The development of a high-quality, open-source environment and a promising baseline policy is a valuable resource for further research in this area.
- [Originality]: The integration of structural correspondence (GAM) with diffusion-based trajectory generation (SADP) for category-level generalization is novel.

Weaknesses:
The paper's primary weakness stems from a significant mismatch between the scope of its claimed contributions and the focus of its empirical evaluation. The authors frame the work as having three main contributions: the DexGarmentLab environment, the automated data collection pipeline, and the HALO policy. However, the experimental section is almost exclusively dedicated to evaluating the performance of the HALO policy against baselines and ablations. The other two pillars of the contribution lack corresponding empirical validation, creating a structural imbalance in the paper.
- [Lack of Experimental Validation for DexGarmentLab]: While the qualitative comparisons in Table 1 and the descriptive improvements to the physics model are informative, they are not supported by experiments. While a key claim is that these new physical parameters reduce the sim-to-real gap and provide more realistic interactions, there is no experiment designed to quantify this benefit. For instance, an analysis showing that training a policy is more sample-efficient or faster in terms of wall-clock time compared to other environments or using the AttachmentBlock method would be necessary to empirically substantiate the environment's value as a standalone scientific contribution.
- [Lack of Quantitative Analysis for the Data Collection Pipeline]: Similarly, the automated data collection pipeline is also presented as a major contribution. However, the quality of the data it produces is never directly measured or analyzed. The paper provides no evidence to show that the generated demonstrations are indeed "expert" or how they compare to demonstrations collected via other methods, such as human teleoperation. The value is thus only implicitly suggested through the HALO policy trained on its data.

**Note**: I acknowledge that this paper possesses a significant amount of effort. The above weaknesses only apply **if the intention is to present the contribution as a unified whole.** Alternatively, and perhaps more effectively, the paper's narrative could be reframed. The work would present as a more coherent and defensible contribution if HALO were positioned as the primary focus, with the environment and data collection pipeline framed as a novel and sophisticated infrastructure built to enable this core research. This would still be a strong contribution, and I will still maintain positive if the authors prefer reframing.

- [Minor]: Citation for PBD is missing: Müller, Matthias, Bruno Heidelberger, Marcus Hennix, and John Ratcliff. "Position based dynamics." Journal of Visual Communication and Image Representation 18, no. 2 (2007): 109-118.

---

> ### Author Rebuttal · Authors · 2025-07-30
>
> # Rebuttal for ntYt
> Thanks for your helpful and meaningful reviews! We will provide a detailed response below to the issues outlined in the Weakness and Question sections.
>
> ---
>
> ## **Weakness:**
>
> > [Lack of Experimental Validation for DexGarmentLab]: ......an analysis showing that training a policy is more sample-efficient or faster in terms of wall-clock time compared to other environments or using the AttachmentBlock method would be necessary to empirically substantiate the environment's value as a standalone scientific contribution.
>
> > [Lack of Quantitative Analysis for the Data Collection Pipeline]: ......the quality of the data it produces is never directly measured or analyzed. The paper provides no evidence to show that the generated demonstrations are indeed "expert" or how they compare to demonstrations collected via other methods, such as human teleoperation.......
>
> ## **Answer:**
>
> Thanks a lot for pointing out these issues! They are highly valuable and constructive suggestions for improving DexGarmentLab. Below, we provide additional clarification regarding the missing empirical validation in the **Environment** and **Data Collection Pipeline** components.
>
> 1.As for **Lack of Experimental Validation for DexGarmentLab**:
>
> Firstly, it is important to clarify that garment simulation in our work is built upon NVIDIA PhysX's Position-Based Dynamics (PBD) framework, which currently offers the most realistic and stable performance for garment simulation. **Alternative simulators such as SofGym or PyBullet do not support PBD-based simulation and generally exhibit inferior performance in soft body dynamics. Moreover, they also lack support for fine-grained interactions between dexterous hands and garments.** Therefore, in terms of garment simulation fidelity and interaction capability, Isaac Sim remains the most suitable and advanced platform available to date.
>
> Currently, the two main open-source garment simulation platforms based on Isaac Sim are GarmentLab and DexGarmentLab. GarmentLab adopts an **attach block-based approach** for garment manipulation, where grasping is achieved by explicitly attaching or detaching blocks to the garment. However, this method **has significant limitations.** Specifically, **attachment and detachment must be explicitly triggered in code, which makes it unsuitable for end-to-end or reinforcement learning-based policy training.** Instead, it is more appropriate for hierarchical planning or hardcoded control, such as those guided by vision-language models (VLMs).
>
> Moreover, **as discussed in the main paper (Lines 130–132), even minimal contact—such as a single finger block touching the garment—can result in attachment and lifting, which is clearly unrealistic.** In the early stages of our project, we experimented with training policies using the attach block method. However, during evaluation, the instability of the attach/detach mechanism often caused the garment to behave erratically or "fly off," leading to frequent task failures. As a result, we ultimately abandoned this approach.
>
> **To further illustrate this point, we selected two representative tasks to evaluate the performance of the attach block-based method.** The results clearly show that this approach performs poorly across both tasks.
>
> | Typical Task Name | Fling Dress (Simulation) | Hang Coat (Simulation) |
> |:---:|:---:|:---:|
> HALO (AttachBlock)  |0.06±0.01|0.15±0.03|
> HALO (Ours)    |**0.82±0.06**|**0.90±0.01**|
>
> 2.As for **Lack of Quantitative Analysis for the Data Collection Pipeline**:
>
> This missing component indeed affects the overall structure of the paper. To address this, we have provided the relevant quantitative analysis below.
>
> **We use LeapMotion for teleoperation data collection in simulation and an exoskeleton device for teleoperation in the real world.** Three representative tasks are selected for data collection, training, and evaluation. The performance of teleoperated policies is then compared with that of policies trained on autonomously collected data. The results are presented below.
>
>
> | Task Name | Hang Tops (Simulation) | Wear Bowlhat (Simulation) | Fold Tops (Real_World)
> |:---:|:---:|:---:|:---:|
> HALO (Automated_Collection_Data) | 0.92±0.04 | 0.72±0.04 | 13 / 15 |
> HALO (TeleOperation_Data)| 0.88±0.03 | 0.70±0.06 | 13 / 15 |
>
> Experimental results indicate that **policies trained on teleoperated and automated data achieve comparable performance** during evaluation. However, from the perspective of human and time cost, teleoperation-based data collection is significantly more labor-intensive and time-consuming than automated data collection. It is also important to note that our data is not synthetically generated, but collected through real executions in either simulation or the real world, which means the demonstrations are physically realistic. The entire data collection pipeline is enabled by model inference and one-shot demonstrations.
>
> 3.As for **summary**:
>
> Thanks again for your valuable suggestions regarding the organization and presentation of the paper. To be honest, we did encounter considerable difficulty when structuring the overall narrative of the paper. Receiving such constructive feedback during the rebuttal stage has been truly helpful for us. **We will sincerely incorporate your suggestions in the revised version, including adding the missing content (such as citations for PBD) and further refining the structure and clarity of DexGarmentLab.** Thank you once again!
>
> ---
>
> ## **Question1：**
>
>  > The entire data collection process is bootstrapped from a single, manually defined expert demonstration. How does the system perform if this initial demonstration is varied? For instance, if you provide a slightly different (but still successful) folding sequence, how does that affect the diversity and quality of the 100 generated demonstrations and the performance of the resulting policy?
>
> ## **Answer:**
>
> To address this question, we conducted additional experiments on the **Fold Tops** task by providing two new variations of initial demonstrations:
>
>  1）**First Variation**: The task sequence remains unchanged (i.e., folding the left sleeve to the right collar, the right sleeve to the left collar, then folding the bottom hem to the collar), but the demo points used to locate key regions are altered.
>
>  2）**Second Variation**: The task sequence itself is modified—first folding the left sleeve to the right hem, then the right sleeve to the left hem, followed by folding both hems toward the collar. The demo points for keypoint localization are also different from the original.
>
> For each case, we collected 100 demonstrations and trained the HALO model accordingly. The evaluation results are as follows:
>
>  | Demo Type | Original Demo | Demo Variant 1 | Demo Variant 2
> |:---:|:---:|:---:|:---:|
> HALO | 0.81±0.03 | 0.84±0.02 | 0.78±0.02 |
>
> From the evaluation results, models trained on demonstrations collected with different task sequences perform comparably during validation. This suggests that changing the task sequence does not significantly affect the quality or diversity of the demonstrations, nor does it have a notable impact on the final policy performance.
>
> ---
>
> ## **Question2-1：**
>
> > Could the authors comment on the extensibility of DexGarmentLab to facilitate future research? For example, what is the workflow for tuning the simulation parameters if a new garment is introduced? The paper mentions parameters were "carefully tuned"; is there a systematic procedure for this, or does each new asset require extensive, ad-hoc manual tuning?
>
> ## **Answer:**
>
> **Our simulation parameters are carefully tuned for garment manipulation and are suitable for most tasks without the need of further adjustment.**
>
> For researchers seeking customization, we will provide a markdown document in the GitHub repository detailing garment-related parameters, including official references and effect descriptions. Additionally, key parameters in **Env_Config/Garment/Particle(Deformable)_Garment.py** are marked with '**# important**' to help researchers focus on critical settings.
>
> ---
>
> ## **Question2-2：**
>
> > How readily can the policy and data collection frameworks be adapted to different hardware? How hard would it be to map the 60-DoF action space and the LeapMotion data collection to a new bimanual setup with different kinematics?
>
> ## **Answer:**
>
> From an implementation perspective, extending the system to a new bimanual setup involves the following steps:
>
> **1）Motion Control in Simulation and Real World:**
>
> In simulation, arm motion control can be implemented using libraries such as **Curobo**, while hand control can be adapted based on our provided codebase.
>
> In the real world, most commercially available arms and hands come with well-encapsulated motion control interfaces, which can be directly used.
>
> **2）Customizing Hand Poses:**
>
> If the hand has a low number of degrees of freedom (DoF), the joint configurations can be manually specified without relying on teleoperation.
>
> For hands with higher DoF, teleoperation devices such as LeapMotion, VR systems, or manufacturer-provided interfaces are recommended. Several open-source repositories for hand teleoperation are also available and have reached a high level of maturity. For LeapMotion, users can refer to our codebase and modify it to accommodate different dexterous hand configurations.
>
>
> **3）Note:**
>
> We do not recommend directly mapping the 60-DoF action space and the LeapMotion-based data collection to a new bimanual setup with different kinematics.
> This is because not only the action space differs, but the visual observations also vary.
> To collect data and train policies under a new bimanual setup, it is necessary to rebuild the simulation and real-world environment, and adapt both joint states and action spaces accordingly, in line with the kinematic structure of the new robotic system.

---

> > ### Comment · Reviewer_ntYt · 2025-08-01
> > **Post-rebuttal**
> >
> > I appreciate the detailed response from the authors. My concerns are well addressed. I will raise my score to accept.

---

### Official Review · Reviewer_MdvE · 2025-06-27

**Clarity:** 2
**Significance:** 3
**Originality:** 3
**Rating:** 4
**Confidence:** 4

**Summary:**

The paper presents DexGarmentLab, a data generation pipeline for garment manipulation and a generalizable manipulation policy using dexterous hands. To eliminate the extensive human effort typically required in imitation and reinforcement learning, the authors propose an automated data collection strategy that enables the generation of large-scale data from a single expert demonstration. An affordance model is employed to transfer manipulation strategies from a source garment to diverse target garments. Leveraging the generated dataset, the authors introduce a generalizable policy framework capable of performing garment manipulation across different scenarios. Extensive experiments are conducted to validate both the effectiveness of the data generation process and the generalizability of the learned policy. Real-world evaluations further demonstrate the practical applicability of the proposed approach.

**Questions:**

- Why could the authors claim their GAM as a "generalizable policy"? Using which mechanisim could the authors truly guarantee the generalization ability or "ensure the generalization for xxx" (line 217)? Except for experiments conducted on cloth with different materials and deformations, the paper does not provide deeper analysis—either from a theoretical perspective or through more fundamental insights—to support the generalization claim. I understand that the method is developed to improve the generalizability. In this case, it would be better not to use such a strong tone since the method cannot fully gaurantee 100% generalization ability.
- How to generate plausible hand grasping poses in the automated data collection process? The authors mentioned LeapMotion in 4.2. However, after checking the appendix for details, LeapMotion seems to be a teleoperation interface that transfers human hand poses to the shadow hand poses. If the data collection is a teleoperation, then everything is reasonable. But it is claimed as an "automated" one.  So what's the automatic grasping pose generation process using LeapMotion? Besides, how to make sure that the grasping pose is physically valid? Would a filtering mechanism be introduced to ensure the plausibility and task completion? If so, what about the success rate of the demonstration generation?
- Real-world evaluations are conducted on a Psi Bot with dexterous hands, which are different from the shadow hand. So, how to directly transfer the policy trained on shadow hands to the real robot? Would it require retraining the policy using Psi Bot in the simulator? If so, we need to re-collect demonstrations -- and since the LeapMotion is designed for shadow hand -- how to generate grasping poses for Psi Bot?

**Ethical Concerns:**

["NO or VERY MINOR ethics concerns only"]

**Final Justification:**

DexGarment focuses on dexterous garment manipulation. The method, including a data generation pipeline and the policy design, is technically sound. Thorough experiments conducted in the simulated environment and the real world further validate the effectiveness of the proposed technique. It makes a solid contribution to the community and merits acceptance.

**Limitations:**

- Insufficient comparisons. The paper lacks comparisons with reinforcement learning (RL)-based approaches and imitation learning (IL) strategies that use demonstrations collected via teleoperation. Without these baselines, it is difficult to fully assess the effectiveness and advantages of the proposed demonstration generation method and policy training strategy.
- Presentation quality. (1) Sections 2 and 3 occupy a disproportionate amount of space relative to their content. (2) Some claims throughout the paper are stated too strongly, which may overstate the method's contributions. (3) Several figures could be made more concise to enhance clarity and improve the overall flow.
- Restricted scientific value. (This is a relatively minor point and has little impact on my overall rating, but I include it for completeness.) The technical innovations and the core message of the paper do not appear to offer fundamentally new insights or contributions to the field.

Overall, DexGarmentLab proposes a promising manipulation environment tailored for a range of tasks involving deformable garments. The thoughtful design of the simulation setup, task environment, garment assets, and the proposed policy training strategy collectively contributes a useful resource to the community. So I would like to give a positive rating. Considering limitations like insufficient experimental comparisons, issues with presentation quality, and a relatively modest level of scientific value, my support would be limited to BA.

**Quality:**

3

**Strengths And Weaknesses:**

Strengths

- Good motivations and valuable insights. The authors propose an efficient and automatic garment manipulation data collection strategy where only one expert demonstration is required, based on the observation that leveraging RL or IL to collect data would be quite time-consuming and human-labor intensive.
- Well-designed method. The data generation pipeline effectively leverages affordance modeling to transfer manipulation strategies from one object to new objects. The policy designed to achieve generalization across different garments is also reasonably designed.

Weakness

- The task difficulty is restricted. Although the task involves long-horizon cloth folding using dexterous hands. The manipulation challenge is not that high --- such manipulation could also be completed by two grippers. The value of dexterous hands, as well as their superiority over parallel grippers, are not clearly demonstrated or justified in the context of the garment manipulation tasks considered in this work.
- Experiments lack a comprehensive comparison with RL-based approaches. Although RL requires reward tuning, there are many works that have explored how to leverage LLMs and VLMs to accelerate the reward tuning process. Compared to IL with demonstrations generated by the proposed approach or collected via teleoperation, RL policies can automatically explore diverse solutions to complete the task and would be much more robust. It would be better not to exclude such types of baselines.
- Section 3 takes too much space -- more than 4 pages, which leaves quite insufficient space for the experiment section. The method section and the related work section could be made more concise than its current version.

---

> ### Author Rebuttal · Authors · 2025-07-31
>
> # Rebuttal for MdvE
> Thanks for your helpful and meaningful reviews! We will provide a detailed response below to the issues in the Weakness and Question sections.
>
> ---
>
> ## **Weakness1:**
>
> > The tasks are limited in difficulty, and the advantages of dexterous hands over parallel grippers are not clearly demonstrated.
>
> ## **Answer:**
>
> Due to current restrictions in simulation technology, certain complex tasks cannot yet be effectively implemented in simulated environments. However, across the existing tasks, dexterous hands still demonstrate clear advantages over parallel grippers in several aspects.
>
> 1.Our real-world experiments highlight **the limitations of two-finger grippers in handling thin or flat garment regions(such as), where stable grasps often fail due to slipping or missed contacts**. Many existing demonstrations rely on grasping the garment’s boundary, but these strategies also often leads to repeated failed attempts. In contrast, our system employing a five-finger dexterous hand demonstrates clear advantages. **Owing to the hand’s larger operation range, greater surface contact, and coordinated multi-finger control, it can stably grasp various regions of a garment—not just its edges—and maintain a firm hold throughout the manipulation process.** This capability greatly enhances the flexibility and robustness of robotic garment handling, freeing the system from constraints on graspable areas or predefined strategies.
>
> 2.Furthermore, **the diverse hand postures offered by the dexterous hand allow us to tailor the manipulation strategy to the specific requirements of each task.**
> - to accomplish a fine-grained task such as putting on a glove, the hand can adopt a **pinch** posture to precisely grasp the glove edge.
> - For stable garment pick-and-place operations, an **open/close** posture can be used to maximize contact and grip.
> - When placing folded clothes, a **cradle** posture helps preserve the folded structure.
> - To smooth wrinkles, the hand can switch to a **smoothening** pose that maximizes surface coverage.
>
> These example poses are shown in the bottom-left corner of Figure 1 (main paper).
>
> The dexterous hand not only replicates the functionality of two-finger grippers but also enables finer and more robust manipulation of garments.
>
> 3.Although current simulation still faces restrictions in precisely modeling highly complex interactions and multi contact between deformable objects and dexterous hands, these limitations would be gradually addressed with the continued advancement of computer graphics and robotics technologies, just like Sapien and Isaac Sim. As a result, **the advantages of dexterous hands will become even more prominent in complex tasks such as assistive dressing or knot tying, which require flexible coordination among multiple fingers.** We will continue to extend tasks and optimize the simulation framework based on DexGarmentLab, aiming to support a wider range of more complex dexterous-hand garment interaction tasks.
>
> ---
>
> ## **Weakness2:**
>
> >  The paper lacks comparisons with RL and IL methods based on teleoperated demonstrations.
>
> ## **Answer:**
>
> We have incorporated all reviewer suggestions related to the benchmark and additionally included four new baselines for comparison: **ACT (IL), pi0 (VLA), RDT (VLA), and Eureka (RL+VLM)**:
>
> | Typical Task Name | Fling Dress (Simulation) | Fold Trousers (Simulation) | Hang Coat (Simulation) | Wear Bowlhat (Simulation) | Fold Tops (Real World) |
> |:---:|:---:|:---:|:---:|:---:|:---:|
> ACT (IL) |0.35±0.02|0.49±0.06|0.43±0.04|0.51±0.03|7/15|
> pi0 (VLA) |*0.69±0.01*|0.52±0.06|*0.72±0.01*|*0.59±0.02*|10/15|
> RDT (VLA) |0.60±0.02|*0.58±0.02*|0.62±0.02|0.48±0.01|9/15|
> Eureka (RL+VLM) | / | / |0.16±0.03|0.08±0.02| / |
> HALO (Ours)    |**0.82±0.06**|**0.77±0.02**|**0.90±0.01**|**0.72±0.04**|**13/15**|
>
> Analysis for **RL-Based Methods**:
>
> We selected Eureka as the representative RL-VLM-combined method and conducted preliminary experiments on several tasks in simulation. While we observed some initial successful cases on simpler tasks such as Hang Coat and Wear Bowlhat, the overall success rate remained low. Additionally, due to the lack of robust parallelization support in the current Isaac Sim environment and the limited effectiveness of reward functions in handling long-horizon tasks like Fling and Fold, this baseline did not yield valid results across all tasks. Nevertheless, we are actively developing a multi-parallel deformable object manipulation environment based on Isaac Lab, and RL-based approaches for deformable manipulation will be a key focus of our future work.
>
> **Comparison with teleoperation data**:
>
> **For teleoperation, We use LeapMotion in simulation and an exoskeleton device in the real world.** The results are presented below.
>
>
> | Task Name | Hang Tops (Simulation) | Wear Bowlhat (Simulation) | Fold Tops (Real_World)
> |:---:|:---:|:---:|:---:|
> HALO (Automated_Collected) | 0.92±0.04 | 0.72±0.04 | 13 / 15 |
> HALO (TeleOperated)| 0.88±0.03 | 0.70±0.06 | 13 / 15 |
>
> Experimental results indicate that **policies trained on teleoperated and automated data achieve comparable performance** during evaluation.
>
> ---
>
> ## **Weakness3 & Question1:**
>
> > Section 3 takes too much space......
>
> > Why could the authors claim their GAM as a "generalizable policy"?......
>
> ## **Answer:**
>
> Thanks a lot for your quite helpful advice on paper organization and presentation! **We believe these suggestions are highly valuable for DexGarmentLab and we will address these problems in camera-ready version to further enhance the quality of the paper.**
>
> ---
>
> ## **Question2:**
>
> > ...... explain details on how LeapMotion generates grasping poses, how physical validity is ensured, whether a filtering mechanism is used, and the success rate of demonstration generation.
>
> ## **Answer:**
>
> For single specific task category (e.g., Hang Coat), the hand poses required to perform specific actions in specific regions, can generally be treated as consistent. Moreover, the deformable nature of garments allows them to adapt naturally to the dexterous hand's pose during manipulation. Therefore, for each specific task, we first define a set of task-specific hand poses (the number depends on the task; for example, Hang Coat requires both a closed grasping pose for the collar and an open pose for release). These poses are generated via teleoperation using LeapMotion, with physical feasibility considered during generation. In this way, the pose definition process inherently includes a manual filtering step to select the most suitable poses.
>
> These task-specific hand poses are then transferred directly into the data collection process for that task category. That is, during data collection across different garments within the same task, the same predefined hand poses are reused.
>
> To further validate the effectiveness of our automated data collection pipeline, we report the success rates for all tasks in the table below.
>
> | Task Name | Fling Dress | Fling Tops | Fling Trousers | Fold Dress | Fold Tops | Fold Trousers | Wear BowlHat |
> |:---:|:---:|:---:|:---:|:---:|:---:|:---:|:---:|
> | Collection Success Rate |92.6%（100/108）|90.1%（100/111）|90.1%（100/111）|84.7%（100/118）|82.0%（100/122）|82.6%（100/121）|95.2%（100/105）|
>
> | Task Name | Hang Coats |Hang Dress | Hang Tops | Hang Trousers | Store Tops |  Wear BaseballCap | Wear Scarf |
> |:---:|:---:|:---:|:---:|:---:|:---:|:---:|:---:|
> | Collection Success Rate |92.6%（100/108）|90.9%（100/110）|91.7%（100/109）|99%（100/101）|98.0%（100/102）|85.5%（100/117）|83.3%（100/120）|
>
> | Task Name | Fold Tops (Real_World) |Hang Tops (Real_World) | Wear Scarf (Real_World) | Wear_Hat (Real World) |
> |:---:|:---:|:---:|:---:|:---:|
> | Collection Success Rate |90.9%（50/55）|96.2%（50/51）|83.3%（50/60）|93.8%（50/53）|
> | Single Collection Time |0min50s|0min38s|1min15s|0min36s|
>
> ---
>
> ## **Question3:**
>
> > ......How to directly transfer the policy trained on shadow hands to the real robot? Would it require retraining the policy using Psi Bot in the simulator? ......the LeapMotion is designed for shadow hand -- how to generate grasping poses for Psi Bot?
>
> ## **Answer:**
>
> To prevent potential misunderstandings, we clarify this point in advance: the real-world experiments presented in the submitted paper are not the result of a policy-level sim-to-real transfer.
>
> Actually, there are two ways using HALO for garment manipulation tasks:
> 1) **transfer the data collection pipeline from simulation to real world, conduct automated data collection and policy training directly in the real world.**
> 2) **train the policy in the simulation and transfer the policy to the real world.**
>
> In our paper, we adopt **way 1** for demonstrate the effectiveness of HALO. In the section *Question2*, we report the efficiency and success rate of real-world data collection, highlighting the strong sim-to-real performance of GAM.
>
> Sim-to-real transfer from the simulation environment remains an important aspect of our study. To this end, **we aligned the settings of both the simulation and real-world environments by using the same hardware setup—Shadow Hand and UR10e—and selected two tasks, Hang Trousers and Wear Hat, for policy-level sim-to-real transfer, which means way 2.** The evaluation criteria follow those used in the previous real-world experiments.
>
> | Task Name | Only Simulation Data | Simulation Data + 15 Real-World Data |
> |:---:|:---:|:---:|
> Hang Trousers |  8 / 15 (53.3%) | 13 / 15 (86.7%) |
> Wear Hat| 9 / 15 (60.0%) | 13 / 15 (86.7%) |
>
> So if you want to train the policy in simulation and use a sim-to-real method to transfer policy, you need to align the settings of both the simulation and real-world environments, in the meanwhile, re-collect demonstrations for training. As for psibot setting, because psibot G0-R dexhand only has six freedoms, we can just define each freedom by ourselves to generate target hand pose, or we can also use teleoperation device in real world to get target pose.

---

> ### Comment · Reviewer_MdvE · 2025-08-03
>
> Thanks for the detailed rebuttal. My questions regarding the real-world experiment on PsiBot, the use of LeapMotion for data collection, and the comparison with IL/RL methods have been properly addressed. Overall, I believe this is a solid paper. With its rigorous study on dexterous garment manipulation and comprehensive benchmarking, the work makes a valuable contribution to the community. I will maintain my positive rating.
> The reason I am not increasing the score is that, as an experiment-oriented application paper, it does not convincingly demonstrate the ability to tackle more challenging garment manipulation tasks or show improved generalization compared to prior works. Pi series has demonstrated quite impressive garment manipulation results, though using grippers. According to the real-world comparisons presented in the rebuttal, pi0 performs worse than HALO -- but I think perhaps it is an unfair comparison. Based on the videos available on the project website, the Fold Tops task does not appear particularly difficult or long-horizon. It involves grasping the garment from a table and hanging it—something that should still be feasible with grippers. I cannot understand why pi0 shows worse performance --- they have demonstrated impressive generalization ability and the ability to tackle long-horizon tasks. If the data is enough, this task would pose a significant difficulty.

---

### Official Review · Reviewer_b8Ya · 2025-06-30

**Clarity:** 3
**Significance:** 3
**Originality:** 3
**Rating:** 5
**Confidence:** 4

**Summary:**

This paper is dedicated to solving the dexterous garment manipulation problem. It introduces a new simulation environment (DexGarmentLab), an effective method for automatically collecting demonstration data from a single expert example, and a hierarchical policy (HALO) that generalizes across diverse garments. Extensive experiments in both simulation and the real world validate the approach, showing it significantly outperforms existing methods.

**Questions:**

see the weaknesses

**Ethical Concerns:**

["NO or VERY MINOR ethics concerns only"]

**Final Justification:**

The paper presents a complete and well-structured framework for dexterous manipulation, featuring a large-scale simulated environment (DexGarmentLab) with diverse tasks, an automated data pipeline that generates varied demonstrations from single examples, and a novel policy. The method is technically sound, with clear ablation studies validating key design choices. Experiments on simulated and real-world tasks demonstrate significant performance improvements over baselines, showcasing the framework's effectiveness and real-world applicability.

**Limitations:**

yes

**Quality:**

3

**Strengths And Weaknesses:**

**Strengths**

- The paper delivers a complete framework—a novel simulated environment, an automated data pipeline, and a new dexterous garment manipulation policy—creating a valuable resource that can accelerate research in this field.
- The proposed DexGarmentLab is a novel and large-scale simulation environment, which features over 2,500 garments, 15 task scenarios.
- The proposed automated data collection method uses garment structure to generate diverse demonstrations from a single example, which effectively solves a major bottleneck for imitation learning.
- The method's effectiveness is convincingly demonstrated through rigorous experiments on 14 simulated and 4 real-world tasks, with clear ablation studies and significant improvements over some baselines.


**Weaknesses**

- The methodological comparison is limited to Diffusion Policy (DP) and 3D Diffusion Policy (DP3). Including other relevant imitation learning or deformable object manipulation methods would provide a more comprehensive performance benchmark.
- The quality of the automatically generated data is not quantitatively compared against traditional data collection methods (e.g., human teleoperation), which could further validate the pipeline's effectiveness. Moreover, the paper emphasizes the efficiency of the automated data collection pipeline. What is the success rate of the pipeline in generating valid and successful task demonstrations that can be used for policy training?
- A more in-depth analysis of the sim-to-real transfer, including specific challenges and the impact of perception noise in the real world, would provide greater insight.
- The paper would benefit from a clearer discussion on the boundaries of the policy's generalization, including an analysis of specific failure cases on more complex or unusual garments.

---

> ### Author Rebuttal · Authors · 2025-07-31
>
> # Rebuttal for b8Ya
> Thanks for your helpful and meaningful reviews! We will provide a detailed response below to the issues outlined in the Weakness and Question sections.
>
> ---
>
> ## **Weakness1:**
>
> > The methodological comparison is limited to Diffusion Policy (DP) and 3D Diffusion Policy (DP3). Including other relevant imitation learning or deformable object manipulation methods would provide a more comprehensive performance benchmark.
>
> ## **Answer：**
>
> We have incorporated all reviewer suggestions related to the benchmark and additionally included four new baselines for comparison: **ACT (IL), pi0 (VLA), RDT (VLA), and Eureka (RL+VLM)**. We selected representative tasks for evaluation, including **Fling Dress, Fold Trousers, Hang Coat, and Wear Bowlhat in simulation, as well as Fold Tops in real world**. The evaluation protocol remains consistent with the original paper:
> - For simulation tasks, each task is evaluated over 50 episodes using three different random seeds. We report the success rates as Mean ± Std across all trials.
> - For real-world task, we evaluate using 3 distinct garments per category, each tested under 5 different initial deformations. We report the success rates as successful_trials / all_trials.
>
> | Typical Task Name | Fling Dress (Simulation) | Fold Trousers (Simulation) | Hang Coat (Simulation) | Wear Bowlhat (Simulation) | Fold Tops (Real World) |
> |:---:|:---:|:---:|:---:|:---:|:---:|
> DP (IL)  |0.59±0.05|0.47±0.04|0.52±0.04|0.41±0.05|*9/15*|
> DP3 (IL) |0.51±0.03|0.54±0.07|0.58±0.04|0.55±0.04|8/15|
> ACT (IL) |0.35±0.02|0.49±0.06|0.43±0.04|0.51±0.03|7/15|
> pi0 (VLA) |*0.69±0.01*|0.52±0.06|*0.72±0.01*|*0.59±0.02*|10/15|
> RDT (VLA) |0.60±0.02|*0.58±0.02*|0.62±0.02|0.48±0.01|9/15|
> Eureka (RL+VLM) | / | / |0.16±0.03|0.08±0.02| / |
> HALO (Ours)    |**0.82±0.06**|**0.77±0.02**|**0.90±0.01**|**0.72±0.04**|**13/15**|
>
> Analysis For **Imitation-Learning-Based Methods**:
>
> We collect 100 demonstrations for each simulation task and collect 50 demonstrations for each real-world task, which are used for policy training (DP/DP3/ACT). A notable limitation of their performance lies in the inability to accurately grasp the target region of the garment, as well as the lack of fine-grained control during placement based on the garment’s shape(/state).
>
> Analysis for **VLA Methods**:
>
> We combine all the demonstrations of simulation and real-world tasks to fine-tune pi0 and RDT model. In terms of final performance, pretrained models such as pi0 and RDT outperform from-scratch approaches like DP, DP3, and ACT. However, compared to HALO, VLA-based models still exhibit clear limitations in accurately perceiving garment shape and state, and in executing precise grasp-and-place actions, resulting in a noticeable performance gap relative to HALO.
>
> Analysis for **RL-Based Methods**:
>
> We selected Eureka as the representative RL-VLM-combined method and conducted preliminary experiments on several tasks in simulation. While we observed some initial successful cases on simpler tasks such as Hang Coat and Wear Bowlhat, the overall success rate remained low. Additionally, due to the lack of robust parallelization support in the current Isaac Sim environment and the limited effectiveness of reward functions in handling long-horizon tasks like Fling and Fold, this baseline did not yield valid results across all tasks. Nevertheless, we are actively developing a multi-parallel deformable object manipulation environment based on Isaac Lab, and RL-based approaches for deformable manipulation will be a key focus of our future work.
>
> ---
>
> ## **Weakness3:**
>
> > A more in-depth analysis of the sim-to-real transfer, including specific challenges and the impact of perception noise in the real world, would provide greater insight.
>
> ## **Answer：**
>
> To prevent potential misunderstandings, we clarify this point in advance: the real-world experiments presented in the submitted paper are not the result of a policy-level sim-to-real transfer. As can be observed, the real-world setup differs from the simulation environment.
>
> Actually, there are two ways using HALO for garment manipulation tasks:
> 1) **transfer the data collection pipeline from simulation to real world, conduct automated data collection and policy training directly in the real world.**
> 2) **train the policy in the simulation and transfer the policy to the real world.**
>
> In our paper, we adopt **way 1** to demonstrate the effectiveness of HALO. In the section *Weakness2*, we report the efficiency and success rate of real-world data collection, highlighting the strong sim-to-real performance of GAM.
>
> Sim-to-real transfer from the simulation environment remains an important aspect of our study. To this end, **we aligned the settings of both the simulation and real-world environments by using the same hardware setup—Shadow Hand and UR10e—and selected two tasks, Hang Trousers and Wear Hat, for policy-level sim-to-real transfer, which means way 2.** The evaluation criteria follow those used in the previous real-world experiments. It is worth noting that sim-to-real performance is more sensitive to point cloud noise. To address the limited precision of the Realsense D435 in this context, we employed a Kinect camera for more accurate point cloud acquisition.
>
> | Task Name | Only Simulation Data | Simulation Data + 15 Real-World Data |
> |:---:|:---:|:---:|
> Hang Trousers |  8 / 15 (53.3%) | 13 / 15 (86.7%) |
> Wear Hat| 9 / 15 (60.0%) | 13 / 15 (86.7%) |
>
> Experimental results show that due to the gap between simulation and the real world, training the policy solely on simulated data leads to a drop in sim-to-real performance. Incorporating a small amount of real-world data into the training process can effectively enhance the policy’s generalization ability.
>
> ---
>
> ## **Weakness2:**
>
> > The quality of the automatically generated data is not quantitatively compared against traditional data collection methods (e.g., human teleoperation), which could further validate the pipeline's effectiveness. Moreover, the paper emphasizes the efficiency of the automated data collection pipeline. What is the success rate of the pipeline in generating valid and successful task demonstrations that can be used for policy training?
>
> ## **Answer：**
>
> **We use LeapMotion for teleoperation data collection in simulation and an exoskeleton device for teleoperation in the real world.** Three representative tasks are selected for data collection, training, and evaluation. The performance of teleoperated policies is then compared with that of policies trained on autonomously collected data. The results are presented below.
>
>
> | Task Name | Hang Tops (Simulation) | Wear Bowlhat (Simulation) | Fold Tops (Real_World)
> |:---:|:---:|:---:|:---:|
> HALO (Automated_Collection_Data) | 0.92±0.04 | 0.72±0.04 | 13 / 15 |
> HALO (TeleOperation_Data)| 0.88±0.03 | 0.70±0.06 | 13 / 15 |
>
> Experimental results indicate that **policies trained on teleoperated and automated data achieve comparable performance** during evaluation. However, from the perspective of human and time cost, teleoperation-based data collection is significantly more labor-intensive and time-consuming than automated data collection. It is also important to note that our data is not synthetically generated, but collected through real executions in either simulation or the real world, which means the demonstrations are physically realistic.
> The entire data collection pipeline is enabled by model inference and one-shot demonstrations.
>
> The following table reports the data collection time and success rate across all tasks, demonstrating the effectiveness and efficiency of the automated data collection pipeline.
>
> | Task Name | Fling Dress | Fling Tops | Fling Trousers | Fold Dress | Fold Tops | Fold Trousers | Wear BowlHat |
> |:---:|:---:|:---:|:---:|:---:|:---:|:---:|:---:|
> | Collection Success Rate |92.6%（100/108）|90.1%（100/111）|90.1%（100/111）|84.7%（100/118）|82.0%（100/122）|82.6%（100/121）|95.2%（100/105）|
> | Single Collection Time |0min56s|0min55s|0min48s|1min06s|1min08s|0min56s|0min53s|
>
> | Task Name | Hang Coats |Hang Dress | Hang Tops | Hang Trousers | Store Tops |  Wear BaseballCap | Wear Scarf |
> |:---:|:---:|:---:|:---:|:---:|:---:|:---:|:---:|
> | Collection Success Rate |92.6%（100/108）|90.9%（100/110）|91.7%（100/109）|99%（100/101）|98.0%（100/102）|85.5%（100/117）|83.3%（100/120）|
> | Single Collection Time |0min41s|0min48s|0min46s|0min58s|1min05s|0min55s|1min42s|
>
> | Task Name | Fold Tops (Real_World) |Hang Tops (Real_World) | Wear Scarf (Real_World) | Wear_Hat (Real World) |
> |:---:|:---:|:---:|:---:|:---:|
> | Collection Success Rate |90.9%（50/55）|96.2%（50/51）|83.3%（50/60）|93.8%（50/53）|
> | Single Collection Time |0min50s|0min38s|1min15s|0min36s|
>
> ---
>
> ## **Weakness4：**
>
> > The paper would benefit from a clearer discussion on the boundaries of the policy's generalization, including an analysis of specific failure cases on more complex or unusual garments.
>
> ## **Answer：**
>
> Based on both simulation and real-world experimental results, we summarize the following points regarding boundary-related performance:
>
> 1.For garments with occluded or highly deformed regions, the Garment Affordance Model (GAM) does not always predict manipulation points accurately. This can lead to failed or suboptimal grasps by the dexterous hand, impacting the final performance. Nonetheless, GAM performs reliably on most garments. For challenging cases, an initial unfold or fling action is recommended to expose key regions.
>
> 2.The policy struggles with garments that deviate from standard geometries, such as asymmetrical designs (e.g., single-sleeved tops) or heavily adorned costumes, which distort the point cloud and affect both GAM and SADP performance.

---

> > ### Comment · Reviewer_b8Ya · 2025-08-01
> > **post-rebuttal**
> >
> > Thanks to the authors for their efforts during the rebuttal. After carefully reading the responses and comments from other reviewers, all my concerns are well addressed. I will improve my score.

---

### Official Review · Reviewer_bHU9 · 2025-07-02

**Clarity:** 2
**Significance:** 3
**Originality:** 3
**Rating:** 4
**Confidence:** 4

**Summary:**

The authors introduce DexGarmentLab, a novel simulation environment designed for dexterous (especially bimanual) garment manipulation. Built on Isaac Sim, it offers over 2,500 garments across 8 categories and 15 task scenarios with high-quality 3D assets. They also propose an automated data collection pipeline that generates diverse demonstrations using a single expert demonstration, reducing manual intervention. Furthermore, the authors present HALO (Hierarchical gArment manipuLation pOlicy), a generalizable policy that combines affordance points for locating manipulation areas and a diffusion method for generating trajectories. Through extensive experiments, the authors demonstrate that HALO outperforms existing methods in both simulation and real-world settings, showing strong generalization across various garment shapes and deformations.

**Questions:**

See Weaknesses

**Ethical Concerns:**

["NO or VERY MINOR ethics concerns only"]

**Final Justification:**

My concerns have been addressed, so I will maintain my score to support the acceptance of the paper.

**Limitations:**

See Weaknesses

**Paper Formatting Concerns:**

No Paper Formatting Concerns

**Quality:**

3

**Strengths And Weaknesses:**

Strengths：
1. DexGarmentLab Environment: Offers a realistic and diverse simulation platform for garment manipulation tasks.
2. Automated Data Collection: Reduces manual effort by generating diverse demonstrations from a single expert demonstration.
3. HALO Policy: Combines affordance points and a diffusion method to achieve strong generalization across different garment shapes and deformations.
4. Experimental Validation: Demonstrates superior performance and generalization ability of HALO in both simulation and real-world experiments.

Weaknesses：
1. While the authors compare DP and DP3 methods, the study lacks benchmarking against state-of-the-art VLAs (pi0 or RDT) that demonstrate superior performance on folding tasks.
2. While demonstrating performance gains on this benchmark DexGarmentLab, the claimed generalizability of the proposed method requires validation across diverse simulation environments. Additional cross-platform testing would strengthen the evidence for its universal applicability.
3. Could the authors clarify whether this is a single-task or multi-task setting?
4. Please specify the computational costs (time overhead) for both training and inference phases of the algorithm.

---

> ### Author Rebuttal · Authors · 2025-07-31
>
> # Rebuttal for bHU9
> Thanks for your helpful and meaningful reviews! We will provide a detailed response below to the issues outlined in the Weakness and Question sections.
>
> ---
>
> ## **Weakness1:**
>
> > While the authors compare DP and DP3 methods, the study lacks benchmarking against state-of-the-art VLAs (pi0 or RDT) that demonstrate superior performance on folding tasks.
>
> ## **Answer：**
>
> We have incorporated all reviewer suggestions related to the benchmark and additionally included four new baselines for comparison: **ACT (IL), pi0 (VLA), RDT (VLA), and Eureka (RL+VLM)**. We selected representative tasks for evaluation, including **Fling Dress, Fold Trousers, Hang Coat, and Wear Bowlhat in simulation, as well as Fold Tops in real world**. The evaluation protocol remains consistent with the original paper:
> - For simulation tasks, each task is evaluated over 50 episodes using three different random seeds. We report the success rates as Mean ± Std across all trials.
> - For real-world task, we evaluate using 3 distinct garments per category, each tested under 5 different initial deformations. We report the success rates as successful_trials / all_trials.
>
> | Typical Task Name | Fling Dress (Simulation) | Fold Trousers (Simulation) | Hang Coat (Simulation) | Wear Bowlhat (Simulation) | Fold Tops (Real World) |
> |:---:|:---:|:---:|:---:|:---:|:---:|
> DP (IL)  |0.59±0.05|0.47±0.04|0.52±0.04|0.41±0.05|*9/15*|
> DP3 (IL) |0.51±0.03|0.54±0.07|0.58±0.04|0.55±0.04|8/15|
> ACT (IL) |0.35±0.02|0.49±0.06|0.43±0.04|0.51±0.03|7/15|
> pi0 (VLA) |*0.69±0.01*|0.52±0.06|*0.72±0.01*|*0.59±0.02*|10/15|
> RDT (VLA) |0.60±0.02|*0.58±0.02*|0.62±0.02|0.48±0.01|9/15|
> Eureka (RL+VLM) | / | / |0.16±0.03|0.08±0.02| / |
> HALO (Ours)    |**0.82±0.06**|**0.77±0.02**|**0.90±0.01**|**0.72±0.04**|**13/15**|
>
> Analysis For **Imitation-Learning-Based Methods**:
>
> We collect 100 demonstrations for each simulation task and collect 50 demonstrations for each real-world task, which are used for policy training (DP/DP3/ACT). A notable limitation of their performance lies in the inability to accurately grasp the target region of the garment, as well as the lack of fine-grained control during placement based on the garment’s shape(/state).
>
> Analysis for **VLA Methods**:
>
> We combine all the demonstrations of simulation and real-world tasks to fine-tune pi0 and RDT model. In terms of final performance, pretrained models such as pi0 and RDT outperform from-scratch approaches like DP, DP3, and ACT. However, compared to HALO, VLA-based models still exhibit clear limitations in accurately perceiving garment shape and state, and in executing precise grasp-and-place actions, resulting in a noticeable performance gap relative to HALO.
>
> Analysis for **RL-Based Methods**:
>
> We selected Eureka as the representative RL-VLM-combined method and conducted preliminary experiments on several tasks in simulation. While we observed some initial successful cases on simpler tasks such as Hang Coat and Wear Bowlhat, the overall success rate remained low. Additionally, due to the lack of robust parallelization support in the current Isaac Sim environment and the limited effectiveness of reward functions in handling long-horizon tasks like Fling and Fold, this baseline did not yield valid results across all tasks. Nevertheless, we are actively developing a multi-parallel deformable object manipulation environment based on Isaac Lab, and RL-based approaches for deformable manipulation will be a key focus of our future work.
>
> ---
>
> ## **Weakness2：**
>
> > While demonstrating performance gains on this benchmark DexGarmentLab, the claimed generalizability of the proposed method requires validation across diverse simulation environments. Additional cross-platform testing would strengthen the evidence for its universal applicability.
>
> ## **Answer:**
>
> Garment simulation in our work is built upon NVIDIA PhysX's Position-Based Dynamics (PBD) and Finite-Element Method(FEM) framework, which currently offers the most realistic and stable performance for garment simulation. **Alternative simulators such as SofGym or PyBullet only support PBD and generally exhibit inferior performance in soft body dynamics. Moreover, they also lack support for fine-grained interactions between dexterous hands and garments.** Therefore, in terms of garment simulation fidelity and interaction capability, Isaac Sim remains the most suitable and advanced platform available to date, and it's not feasible to perform garment manipulation tasks in simulators outside of Isaac.
>
> However, regardless of the simulation platform used, real-world validation remains essential. Therefore, we directly consider the real-world environment as the final platform. Our real-world garment manipulation experiments demonstrate the effectiveness of our policy in physical settings, further supporting its general applicability.
>
> What's more, **for cross-platform testing, we selected three rigid-object manipulation tasks—covering both single-arm and dual-arm settings—from three different benchmarks: ManiSkill, RLBench, and Robosuite, each built on a distinct simulation backend.** We used DP and pi0 as baselines for comparison, and the success rates were obtained from existing publications.
>
> The results show that HALO performs on par with, or even better than, pi0. This further supports the universal applicability of the HALO strategy across a wide range of tasks, including both rigid and deformable object manipulation.
>
> | Task Name | stackcube (Maniskill) | put_shoe_in_box (RLBench) | two_arm_lifting (Robosuite) |
> |:---:|:---:|:---:|:---:|
> | Backbone | Sapien | CoppeliaSim | Mujoco |
> | DP    | 53.8 % | 42.5%  | 51.0% |
> | pi0   | 92.0% | 78.6% | 86.5% |
> | HALO  | 90.0% | 85.0% | 90.0% |
>
> ---
>
> ## **Weakness3:**
>
> > Could the authors clarify whether this is a single-task or multi-task setting?
>
> ## **Answer:**
>
> If you are referring to whether a single model is trained for a single task or multiple tasks, this is a single-task setting. However, there are also some plausible ways to transfer it to multi-task setting, as discussed in [1,2].
>
> If this is not what you meant, please feel free to let us know and we can have a further discussion.
>
> **ref:**
>
> [1]Hao C, Lin K, Luo S, et al. Language-guided manipulation with diffusion policies and constrained inpainting[J]. arXiv preprint arXiv:2406.09767, 2024.
>
> [2]Ma X, Patidar S, Haughton I, et al. Hierarchical diffusion policy for kinematics-aware multi-task robotic manipulation[C]//Proceedings of the IEEE/CVF Conference on Computer Vision and Pattern Recognition. 2024: 18081-18090.
>
> ---
>
> ## **Weakness4:**
>
> > Please specify the computational costs (time overhead) for both training and inference phases of the algorithm.
>
> ## **Answer:**
>
> The training of **GAM** is performed on a **single RTX 4090 GPU**, requiring approximately **22 GB of GPU memory** when using a batch size of 64. Depending on the dataset size, the training process takes around **12 hours** to complete. GAM requires only about **1 GB of GPU memory** during the inference stage.
>
> The **SADP** module is trained on a **single NVIDIA A800 GPU**, with a memory usage of **approximately 75 GB** when using a batch size of 200. The training process spans approximately **16 hours** for 3000 epochs. However, earlier checkpoints can be selected for validation depending on the use case.
>
> HALO, which integrates both GAM and SADP, requires about **3 GB of GPU memory** during the inference stage. When running HALO inference in the Isaac Sim environment, the total GPU memory footprint is approximately **15 GB**.

---

> > ### Comment · Reviewer_bHU9 · 2025-08-03
> >
> > Thanks for authors' response. My concerns have been addressed, so I will maintain my score to support the acceptance of the paper.

---

### Official Review · Reviewer_JjAQ · 2025-07-03

**Clarity:** 4
**Significance:** 4
**Originality:** 3
**Rating:** 5
**Confidence:** 4

**Summary:**

This paper introduces DexGarmentLab, a novel simulation environment specifically designed for dexterous (especially bimanual) garment manipulation. The environment provides large-scale high-quality 3D 8 assets for 15 task scenarios, and refines simulation techniques tailored for garment 9 modeling to reduce the sim-to-real gap. To address the bottleneck of labor-intensive data collection, the authors propose an automated pipeline that leverages a single expert demonstration and a GAM to generate diverse manipulation trajectories for novel garments. Furthermore, they propose HALO, which combines affordance point inference and a structure-aware diffusion policy to Generate Generalizable Trajectories. Extensive experiments show that the methods outperforms baselines, and generalizing well to unseen garments,shape,deformation,env config.

**Questions:**

1.	Your method uses five-finger dexterous hands instead of common two-finger grippers. What specific advantages do five fingers provide for garment manipulation tasks in your experiments?
2.	Are the grasp points on the garment defined in 2D (on the surface) or 3D coordinates?
3.	Since garments are flexible, did you consider grasping force or pressure to avoid slippage or fabric damage?

**Ethical Concerns:**

["NO or VERY MINOR ethics concerns only"]

**Final Justification:**

Many thanks to authors for their detailed responses. They raised various good points, although I keep my score the same, as it was on the positive side.

**Limitations:**

The authors have adequately addressed the limitations.

**Paper Formatting Concerns:**

No major formatting issues found.

**Quality:**

4

**Strengths And Weaknesses:**

Strengths

1.	The integration of GAM and structure-aware diffusion policy (HALO) is novel and provides a principled way to generalize manipulation across unseen garments and shapes.
2.	The proposed environment and methods are highly relevant for real-world applications in robotics, such as laundry folding, dressing, and assistive robotics.
3.  The figures and tables are of high quality, with clear and informative visualizations that greatly aid understanding.

Weaknesses

1.	In the real-world experiments, particularly on the Fold Tops and Hang Tops tasks, the garments used appear to have relatively simple and uniform colors. It is unclear how well the proposed method would generalize to garments with more complex patterns, prints, or highly varied textures, which are common in real-world scenarios.
2.	In Table 3, please clarify what metric is being reported (e.g., is it the number of successful trials out of total attempts, or another measure?).

---

> ### Author Rebuttal · Authors · 2025-07-30
>
> # Rebuttal for JjAQ
> Thanks for your helpful and meaningful reviews! We will provide a detailed response below to the issues outlined in the Weakness and Question sections.
>
> ---
>
> ## **Weakness1:**
>
> > In the real-world experiments, particularly on the Fold Tops and Hang Tops tasks, the garments used appear to have relatively simple and uniform colors. It is unclear how well the proposed method would generalize to garments with more complex patterns, prints, or highly varied textures, which are common in real-world scenarios.
>
> ## **Answer:**
>
> For our policy training and validation, we **only utilized the XYZ positional coordinates of the point clouds without incorporating color attributes**. This is because, for garments of the same category, color, material, and appearance can vary significantly, while their underlying geometric structure remains largely consistent. Introducing color information could potentially hinder generalization performance by introducing unnecessary variance, as garment colors in the real world are highly diverse and will further be significantly affected by variations in viewpoint and lighting. Since HALO relies solely on positional information of point cloud, it is theoretically robust to variations in color and texture.
>
> To further validate this, **we conducted real-world experiments (Fold Tops) using three distinct types of garments featuring prints, textures and patterns** : (1) those with highly saturated colors, (2) those with complex textures (e.g., black-and-white patterns, color-blocking, or printed designs), and (3) those featuring raised elements (e.g., embroidery or appliqués). For each type, we performed five folding trials, resulting in a total of 15 experiments. **The overall success rate remained at 85% (13/15), confirming the method's robustness.**
>
> However, we acknowledge a limitation: **the policy may underperform when handling garments with atypical structures**, such as asymmetrical designs (e.g., single-sleeved tops) or deviations from conventional garment geometry. While such cases are relatively rare, they represent an important direction for future research.
>
> ---
>
> ## **Weakness2:**
>
> > In Table 3, please clarify what metric is being reported (e.g., is it the number of successful trials out of total attempts, or another measure?).
>
> ## **Answer:**
>
> For real-world evaluation, in every task, we have 3 distinct garments, each with 5 initial deformations and positions, resulting in a total of 15 experiments. **The metric is the number of successful trials out of total 15 experiments.**
>
> ---
>
> ## **Question1:**
>
> > Your method uses five-finger dexterous hands instead of common two-finger grippers. What specific advantages do five fingers provide for garment manipulation tasks in your experiments?
>
> ## **Answer:**
>
> 1.Our experimental results, particularly those real-world results, reveal significant limitations in two-finger grippers when handling thin or flat regions of garments. In such cases, **the gripper frequently fails to achieve a stable grasp, with garments either slipping during manipulation or not being grasped at all**. Many existing demonstrations involve garment folding with two-finger grippers rely on grasping the garment’s boundary, but these strategies also often leads to repeated failed attempts. In contrast, our system employing a five-finger dexterous hand demonstrates clear advantages. **Owing to the hand’s larger operation range, greater surface contact, and coordinated multi-finger control, it can stably grasp various regions of a garment—not just its edges—and maintain a firm hold throughout the manipulation process.** This capability greatly enhances the flexibility and robustness of robotic garment handling, freeing the system from constraints on graspable areas or predefined strategies.
>
> 2.Furthermore, **the diverse hand postures offered by the dexterous hand allow us to tailor the manipulation strategy to the specific requirements of each task.**
> - to accomplish a fine-grained task such as putting on a glove, the hand can adopt a **pinch** posture to precisely grasp the glove edge.
> - For stable garment pick-and-place operations, an **open/close** posture can be used to maximize contact and grip.
> - When placing folded clothes, a **cradle** posture helps preserve the folded structure.
> - To smooth wrinkles, the hand can switch to a **smoothening** pose that maximizes surface coverage.
>
> These example poses are shown in the bottom-left corner of Figure 1 (main paper).
>
> The dexterous hand not only replicates the functionality of two-finger grippers but also enables finer and more robust manipulation of garments.
>
> 3.Although current simulation still faces significant challenges in accurately modeling interactions between deformable objects and dexterous hands—limiting its ability to simulate complex tasks—these limitations are expected to be gradually addressed with the continued advancement of computer graphics and robotics technologies. As a result, **the advantages of dexterous hands will become even more prominent in complex tasks such as assistive dressing or knot tying, which require flexible coordination among multiple fingers.** We will continue to extend tasks and optimize the simulation framework based on DexGarmentLab, aiming to support a wider range of more complex dexterous-hand garment interaction tasks.
>
> ---
>
> ## **Question2:**
>
> > Are the grasp points on the garment defined in 2D (on the surface) or 3D coordinates?
>
> ## **Answer:**
>
> **The grasp points are defined directly in 3D coordinates** .
>
> ---
>
> ## **Question3:**
>
> > Since garments are flexible, did you consider grasping force or pressure to avoid slippage or fabric damage?
>
> ## **Answer:**
>
> This is a highly promising research direction. Current studies in the field are still exploring how to enable robots to grasp garments both accurately and stably. As a result, the most straightforward strategy has been to apply maximum force during grasping to ensure success. However, beyond grasp success, it is equally important to detect garment slippage in real time and to avoid potential fabric damage. This requires considering grasping force or pressure, and leveraging tactile sensing during manipulation. We view this as an important and promising direction for future work. Relevant developments are currently underway in simulation, and we will continue to update and expand this line of research in subsequent work.

---

> > ### Comment · Reviewer_JjAQ · 2025-08-02
> > **Answer to Rebuttal**
> >
> > Thank you to the authors for the clear and thorough responses. All of my concerns have been addressed in the rebuttal. I appreciate the effort and care put into the replies.

---

### Decision · Program_Chairs · 2025-09-17

**Decision:**

Accept (spotlight)

**Comment:**

## Summary

This paper introduces DexGarmentLab, a comprehensive framework for dexterous garment manipulation. The work makes three primary contributions: a new simulation environment with a large dataset of 3D garment assets, an automated data collection pipeline that generates diverse demonstrations from a single expert example, and a novel hierarchical policy (HALO) for generalizable manipulation. The paper is a strong contribution to the field of robotic manipulation and is recommended for acceptance.

## Strength identified by Reviewers
All five reviewers unanimously highlighted the completeness and significance of the framework as a major strength. They praised the integration of a realistic simulation environment, an innovative automated data pipeline, and a novel, effective policy. This complete package is seen as a valuable resource that will accelerate future research in dexterous and deformable object manipulation. The AC believes that the data collection pipeline will be a strong contribution to the research community

## Weaknesses and Rebuttal
The initial reviews raised several constructive weaknesses, which the authors thoroughly addressed in their rebuttal.

- Limited Baselines (bHU9, b8Ya, MdvE): Multiple reviewers pointed out that the initial evaluation was limited, primarily comparing HALO against Diffusion Policy (DP) and 3D Diffusion Policy (DP3).

  - Rebuttal: The authors responded by conducting an extensive new set of experiments, adding four new baselines: ACT (IL), π0 (VLA), RDT (VLA), and Eureka (RL+VLM).

- Lack of Validation for the Environment and Data Pipeline (ntYt): One reviewer noted that there was no quantitative analysis of the data pipeline's quality compared to other methods (e.g., teleoperation) or the simulation environment's benefits.

  - Rebuttal: The authors provided new experiments directly comparing the performance of policies trained on their automatically collected data versus data from human teleoperation. The results showed comparable performance, validating the quality of their pipeline while highlighting its superior efficiency.

- Generalization to Complex Garments (JjAQ): A reviewer questioned how well the method would generalize to garments with more complex patterns, textures, or atypical structures.

  - Rebuttal: The authors clarified that their policy relies on point cloud geometry, not color, making it robust to textures and patterns. They validated this with new real-world experiments on garments with varied prints and textures, achieving a high success rate. They also acknowledged the limitation that the policy may struggle with highly atypical garment structures, defining a clear boundary for the method's current capabilities.

- Clarity on Sim-to-Real Transfer (b8Ya, MdvE): There was some confusion about whether the real-world experiments involved direct sim-to-real policy transfer.

  - Rebuttal: The authors clarified that their primary real-world results came from training on real-world data collected via their automated pipeline. However, they also conducted new policy-level sim-to-real experiments, showing that while there is a performance drop, it can be effectively mitigated by fine-tuning with a small amount of real-world data.

## Recommendation Justification

This paper presents a complete and well-structured framework for dexterous garment manipulation, featuring a large-scale simulated environment (DexGarmentLab), an automated data pipeline that generates varied demonstrations from a single example, and a novel, high-performing policy (HALO). The work is technically sound, with rigorous experiments and ablation studies validating its key design choices on 14 simulated and 4 real-world tasks. The rebuttal led to a clear consensus, with all five reviewers confidently recommending acceptance and some increasing their scores. The resulting work is a significant and high-impact contribution that provides a valuable, open-source resource poised to accelerate future research in the challenging domain of deformable object manipulation.